# Self-Improvement Anomaly Detection via Large Language Model for Unsupervised Zero-shot Anomaly Detection

## Abstract

Zero-shot anomaly detection has emerged to overcome the limitations of conventional methods, which depend on learning the distribution of normal data and struggle to generalize to unseen class. However, existing zero-shot methods rely on anomalous data during training and fail to account for environments where anomalous data are scarce or nonexistent. To address these limitations, we propose a novel unsupervised zero-shot anomaly detection framework, self-improvement anomaly detection with large language model that requires no anomalous data during training. It leverages self-improvement large language model-based architecture that refines textual responses grounded in input images. To support semantic interpretation, we design stage prompts that guide the large language model using visual features spanning from local patterns to global semantics. Our approach not only produces interpretable anomaly maps but also enhances semantic understanding of normality, offering a new direction for zero-shot anomaly detection under realistic anomaly-free constraints. Extensive experiments on nine real-world datasets from both industrial and medical domains demonstrate the effectiveness of our approach. Our self-improvement anomaly detection with large language model outperforms state-of-the-art methods across various unsupervised zero-shot anomaly detection benchmarks, validating its robustness and generalizability across diverse datasets.

## 1 Introduction

Visual anomaly detection is an important task aimed at identifying abnormal or unexpected patterns in various fields such as defect inspection in manufacturing processes (Roth et al., 2022; Hyun et al., 2024; Rudolph et al., 2023; Li et al., 2023a) or the medical imaging diagnosis (Huang et al., 2024; Hua et al., 2024). Traditional approaches have primarily relied on Unsupervised Anomaly Detection (UAD) methods (You et al., 2022; Lu et al., 2023a; Guo et al., 2023), which learn the distribution of normal data and detect deviations from this learned distribution. However, the fundamental limitation of these methods is that they depend exclusively on the distribution of observed normal classes during training, resulting in the limited sensitivity to previously unseen normal variations. To address this limitation, recent advances (Zhou et al., 2023; Cao et al., 2024) in Zero-shot Anomaly Detection (ZAD) have emerged. As illustrated in Fig. 1, ZAD aims to detect anomalies in unseen data by learning both normality and abnormality, often leveraging multimodal representations or large-scale pretrained models. Despite these advances, existing ZAD methods still rely on anomalous data during the training process, which limits their applicability in real-world scenarios where no anomalous samples are available. In this context, training-free ZAD represents a specific subclass that removes the need for explicit training on target data. However, although such training-free approaches bypass the training phase, their heavy reliance on large-scale web-pretrained models often causes domain mismatch, making it difficult to capture rare, domain-specific anomaly patterns such as industrial defects or subtle medical imaging abnormalities.

To overcome the limitations of both frameworks, we propose a novel Unsupervised Zero-shot Anomaly Detection (UZAD), which operates robustly on unseen data without requiring any anomalous samples. As illustrated in Fig. 1, UZAD assumes an anomaly-free training environment in which only normal samples are used during training, while the model is evaluated on both normal

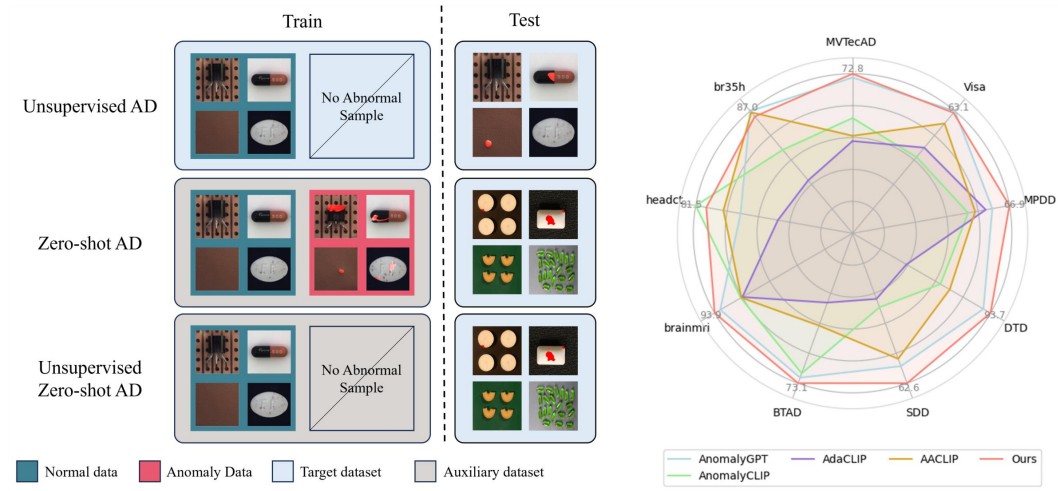

Figure 1: Left: Illustrations for target and auxiliary dataset of unsupervised, zero-shot, and unsupervised zero-shot anomaly detection paradigms. Right: Quantitative comparison with popular methods by image-level AUROC on industrial and medical datasets.

and anomalous samples from previously unseen datasets. This setting more accurately reflects real-world scenarios where anomalous data are scarce or unavailable, and defines a more challenging tasks than conventional ZAD. Unlike existing ZAD approaches, which aim to learn the boundary between normality and abnormality, our UZAD aims to generalize abnormality from the concept of normality. As shown in Figs. 2 (a) and (c), existing ZAD approaches tend to rely on anomalous data during training, which leads to significant performance degradation under the UZAD setting. In addition, as shown in Fig. 2 (b), these approaches often fail to accurately localize anomalies. Considering that UZAD requires learning abnormality without the access to explicitly labeled anomalous samples, auxiliary mechanisms are needed to help the model learn abnormality indirectly. We empirically observed that augmentation-based approaches (Li et al., 2021; Bae et al., 2018), which treat normal images as pseudo-anomalies, can be effective under the UZAD setting. However, augmentation is limited in capturing diverse and semantically abnormal patterns. Accordingly, approaches leveraging Large Language Model (LLM) have gained attention as a promising alternative, offering rich textual expressiveness and advanced text generation capabilities. Recently, several approaches have explored leveraging LLM for visual anomaly detection by integrating visual and textual information. For instance, (Gu et al., 2024) demonstrated the potential of multimodal anomaly detection by utilizing an LLM to generate semantic descriptions of input images. However, this approach also has several limitations. First, they apply a fixed set of predefined text prompt templates uniformly across all images, which restricts flexible context-aware querying or interpretation. Second, since the UZAD framework operates under an anomaly-free training environment, aligning an abnormal prompt with a normal image remains semantically uncertain in relation to actual defects.

To address these limitations, we propose a novel visual anomaly detection framework, Self-Improvement Anomaly Detection with LLM (SIAD-LLM), which leverages image-grounded textual question and answering through a LLM. Our framework primarily utilizes textual responses generated by an LLM given an input image, and embeds the responses into a text encoder to derive semantic textual representations of normality and abnormality. Instead of relying on predefined text prompt templates, the framework dynamically generates context-aware and informative prompts through image-grounded question and answering. This allows not only the detection of anomalies but also enhanced semantic discrimination of normality. The generated textual responses are integrated into the model via the text encoder, refining the internal representations. This process enables the model to dynamically generate and reuse feedback through the self-improvement mechanism.

Additionally, we observe that the features extracted from each stage of the visual encoder capture information ranging from local patterns to global semantics. Based on this observation, we design a novel stage prompt template that integrates stage-wise features into each predefined text prompt.

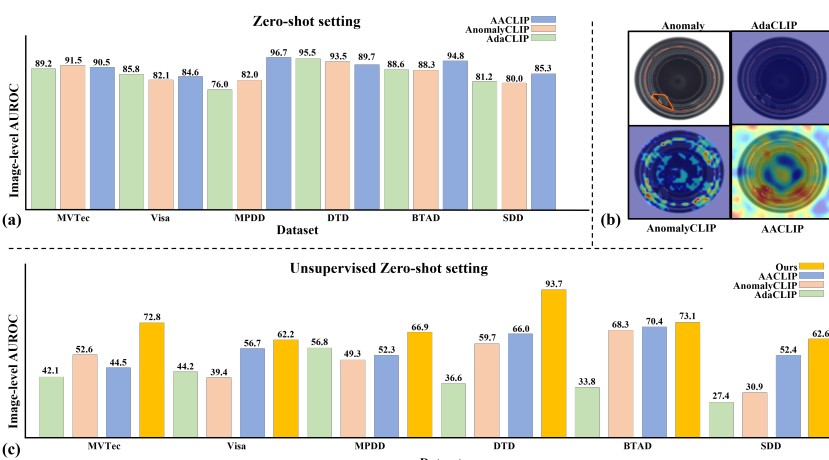

Figure 2: (a) Image-level AUROC of zero-shot methods under the ZAD setting. (b) Qualitative results of zero-shot methods under the UZAD setting. (c) Image-level AUROC for zero-shot methods under the UZAD setting.

This design enables scale-aware anomaly localization and improves semantic discrimination. As shown in Fig. 1, this combination of methods enhances both semantic expressiveness and accurate anomaly localization, and demonstrates robust performance under the UZAD setting. Our key contributions are summarized as follows:

- We present a novel task setting, namely UZAD, to alleviate the challenges of the ZAD problem. UZAD assumes an anomaly-free training environment and evaluates models on unseen datasets containing both normal and anomalous samples.

- We propose SIAD-LLM, a self-improvement anomaly detection framework that leverages image-grounded question and answering with LLM. Instead of relying solely on fixed prompts, the framework generates context-aware responses to improve the expressiveness of textual representations, which helps the model to enhance semantic understanding of normality and abnormality.

- We design a novel stage prompt templates that integrates features extracted from different stages of the encoder, capturing information ranging from local patterns to global semantics. This design improves the model in terms of scale-aware anomaly localization and semantic discrimination capabilities.

## 2 RELATED WORKS

### 2.1 UNSUPERVISED ANOMALY DETECTION

Unsupervised anomaly detection(UAD) (Deng & Li, 2022; He et al., 2024; Lu et al., 2023b; Roth et al., 2022; Cao et al., 2022) aims to learn the normal distribution of a target class using only normal samples, considering real-world scenarios where anomalous samples are scarce or unavailable. Based on the learned normal distribution, the model captures the characteristics of the target class and detects outliers that deviate from this representation. However, these approaches often lack robustness to unseen variations in the normal class distribution, which limits their generalizability in real-world scenarios.

### 2.2 ZERO-SHOT ANOMALY DETECTION

It is a task in which target to detect anomaly in datasets not used during training. Accordingly, it leverages the generalization and zero-shot capability of Vision-Language Model (VLM) such as CLIP (Radford et al., 2021). While VLM are typically trained on large-scale image-text pair datasets, they are not specifically designed for anomaly detection tasks. To bridge this gap, prior

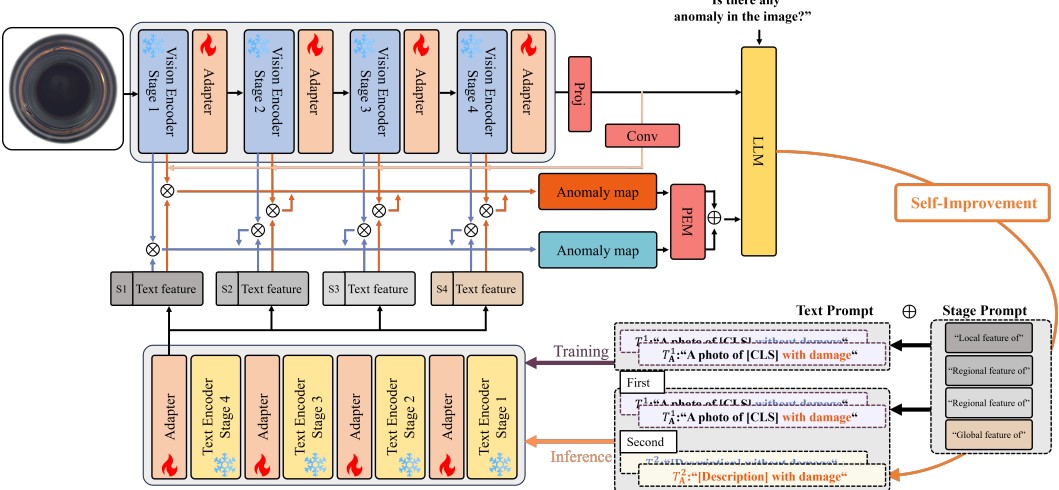

Figure 3: Framework of SIAD-LLM. The model extracts visual features from different stages of the visual encoder, with adapter modules applied at each stage to finetune intermediate representations. Stage-wise features are used to compute anomaly maps. The LLM receives both visual information and prompt queries, then generates responses indicating. These responses are fed back into the model as context-aware prompts, enabling a self-improvement mechanism.

work falls into two families: (i) training-based approaches that learn notions of normality and abnormality from auxiliary data, and (ii) training-free methods that adjust the inference procedure without additional training. AnomalyCLIP (Zhou et al., 2023) learns a single pair of state prompts through object-aware prompt learning. In contrast, AdaCLIP (Cao et al., 2024) utilizes image embeddings as text prompts to dynamically generate appropriate prompts for each input image. However, these methods still rely on anomalous samples to learn representations of abnormality, which limits their applicability in data-restricted real-world scenarios where anomalous data are unavailable for training. Training-free methods like WinCLIP (Jeong et al., 2023) and AnoVL (Deng et al., 2023) craft prompts and modify computation mechanism, but without optimization-based training their performance saturates, especially for pixel-level localization.

## 2.3 LARGE LANGUAGE MODELS IN VISION TASK

LLM (Koroteev, 2021; Touvron et al., 2023; Chiang et al., 2023) have been applied to a wide range of tasks by leveraging their powerful reasoning and generative capabilities. Recently, researchers have extended the reasoning capabilities of LLM to the vision domain, enabling them to process both textual and visual inputs. This advancement has led to the development of Multimodal Large Language Model (MLLM) (Alayrac et al., 2022; Li et al., 2023b; Liu et al., 2023; Zhu et al., 2023; Su et al., 2023). Furthermore, several studies have explored the integration of MLLM into vision-centric tasks. LISA (Lai et al., 2024) feeds the output of an MLLM into a learnable decoder to perform reasoning-based segmentation. DSV-LFS (Karimi & Poullis, 2025) improves segmentation performance by providing class-level descriptions generated by the LLM. In the context of anomaly detection, AnomalyGPT(Gu et al., 2024) extends the application of LLM by enabling the model to detect anomalies and generate responses in the context of anomaly detection.

## 3 METHOD

### 3.1 OVERVIEW

This paper proposes SIAD-LLM, a novel framework that effectively adapts LLM for UZAD. As illustrated in Fig. 3, SIAD-LLM introduces a stage prompt template, allowing each feature representation extracted from different stages of the pretrained text encoder to be independently utilized during

learning. In addition to stage prompts, SIAD-LLM employs text prompt templates to semantically describe each object under both normal and abnormal conditions. For instance, for a given class such as a bottle, text prompts are formulated as `a photo of a bottle with damaged` or `a photo of a bottle without damaged`. These prompts are then tokenized to guide the LLM. To enhance the expressiveness of visual features, we design an Enhancement Module (EM) that improves the quality of the generated anomaly map and maintains semantic consistency between the LLM and the anomaly map. The image embedding and anomaly map are fed into the LLM, which generates a text response that includes a description of the given image. This text response is then fed back to the text encoder, enabling refinement of both the anomaly map and the LLM-generated responses.

## 3.2 UNSUPERVISED ZERO-SHOT ANOMALY DETECTION

We focus on UZAD settings, which aims to detect anomalies from unseen domains or classes at inference phase, using only normal data during the training phase. To address this challenge, pseudo-anomalies are generated through augmentation of normal data, which is then fed into the model along with the original normal samples. Let $\mathcal{X}_{\text{normal}} = \{x_i\}_{i=1}^N$ denote the set of normal training samples.

$$\mathcal{X}_{\text{input}} = [\mathcal{X}_{\text{normal}} \| \{\phi(x) \mid x \in \mathcal{X}_{\text{normal}}\}], \tag{1}$$

$$\mathbf{Z} = \mathcal{F}(\mathcal{X}_{\text{input}}), \tag{2}$$

where $\phi$ is an augmentation operator that generates pseudo-anomalies, and $\mathcal{F}$ denotes a pretrained visual encoder. In our implementation, $\phi$ follows a CutPaste (Li et al., 2021) based strategy that replaces a randomly sampled patch with another patch from a normal image. Poisson smoothing (Pérez et al., 2023) is applied during this process to reduce boundary artifacts and produce more natural pseudo anomalies. The concatenated input $\mathcal{X}_{\text{input}}$ consists of both original normal samples and their augmented pseudo-anomaly counterparts. $\mathbf{Z}$ denotes the output of the pretrained visual encoder. In addition, we apply an adapter module to finetune the pretrained encoder for the anomaly detection task.

## 3.3 STAGE PROMPT TEMPLATE

Traditional vision-language models typically generate text embeddings by uniformly applying a fixed predefined text prompt template across all stages of the visual encoder. However, this approach limits the expressive capacity of the text embeddings, making it difficult to capture fine-grained semantic variations. To address this limitation, we propose a stage prompt template that reflects the characteristics of stage-wise representations by assigning different prompts to each stage. We highlight that the feature representations extracted from different stages of the visual encoder are inherently distinct, as each stage encodes different aspects of the visual input. Specifically, we embed descriptive phrases such as `local feature of`, `regional feature of` and `global feature of` into the prompt structure to capture information ranging from local patterns to global semantics. This design enables the model to leverage complementary information across stages for the same image. Formally, the proposed prompt templates are defined as follows:

$$g_n = [S_k] \, [V_1][V_2] \ldots [V_E] \, [\text{CLS}] \, [\text{without damaged}], \tag{3}$$

$$g_a = [S_k] \, [W_1][W_2] \ldots [W_E] \, [\text{CLS}] \, [\text{with damaged}], \tag{4}$$

where $[V_i]$ and $[W_i]$ $(i = 1, \ldots, E)$ denote the word embeddings for normality and abnormality, respectively. $[S_k]$ $(k = 1, \ldots, 4)$ represents the stage prompt token that aligns with the corresponding encoder stage, and $[\text{CLS}]$ denotes the name of the target category. $g_n$ and $g_a$ refer to the normal and abnormal prompt templates, respectively. These stage prompt templates are concatenated with the base prompt template and used as input to the text encoder. Each prompt is aligned with the feature map extracted from the corresponding stage of the visual encoder, enabling the generation of stage-wise anomaly maps. Finally, the anomaly maps from all stages are aggregated to obtain the final anomaly localization result.

## 3.4 SELF-IMPROVEMENT LLM

The MLLM interprets visual information through image-grounded question and answering and generates corresponding textual responses. However, the direct application of LLM-generated textual

outputs in key components of anomaly detection, such as anomaly scoring and anomaly map generation, has remained unexplored. Existing studies mainly rely on class-level descriptions of fixed predefined text prompts, which do not extract the rich semantic information that LLM can generate. To fill this research gap, we propose a novel self-improvement framework that leverages the outputs of the LLM to refine the textual prompts dynamically. Unlike conventional approaches that treat LLM responses as passive outputs, our method reuses them to iteratively enhance the semantic representations of normality and abnormality. This design allows for more flexible and adaptive learning in anomaly detection. The textual responses generated by the LLM include image-specific descriptions, which are used to augment static prompt templates, serving as context-aware textual prompts. Since these responses are grounded in the visual characteristics of each individual image, they significantly improve the diversity and expressiveness of the prompts. To explicitly indicate normality or abnormality, we append phrases such as `with damaged` or `without damaged` to the generated descriptions. These context-aware prompts are subsequently embedded via a text encoder and processed through the same inference pipeline to obtain the final LLM response. Since the pretrained encoder is trained on large-scale web dataset (Girdhar et al., 2023), embedding the LLM-generated responses back into the model may result in semantic misalignment with the image representations. To alleviate this, we introduce an adapter module that is integrated into the encoder. This adapter is finetuned to align the semantic features derived from the LLM responses with the corresponding visual representations. The adapter process can be formally described as follows:

$$\mathbf{S}'_i = \text{Norm}(\text{LeakyReLU}(\text{Linear}(\mathbf{S}_i))), \quad i = 1, 2, 3, 4 \tag{5}$$

$$\hat{\mathbf{S}}_i = \lambda \cdot \mathbf{S}'_i + (1 - \lambda) \cdot \mathbf{S}_i, \tag{6}$$

where $\mathbf{S}_i$ denotes the feature from the $i$-th stage of the encoder, and $\text{Norm}(\cdot)$ represents $L_2$ normalization. $\lambda$ is a weighting coefficient, which is set to 0.1 in our experiments.

### 3.5 ENHANCED MODULE

#### 3.5.1 STAGE-WISE ENHANCED MODULE

We enhance the contextual information of visual feature representations by integrating the features from the final stage of the visual encoder into all stages. Since the final stage captures global semantic information, we apply convolution and projection layers to the final stage features to adjust the spatial resolution of other stages, then fuse them with the corresponding stage features. The stage-wise enhanced module process can be formally described as follows:

$$\mathbf{S}_{\text{emd}} = \hat{\mathbf{S}}_i + \text{Conv}\left(\text{Proj}(\mathbf{S}^*)\right), \quad i = 1, 2, 3, 4 \tag{7}$$

where $\hat{\mathbf{S}}_i$ denotes the visual feature map at stage $i$, and $\mathbf{S}^*$ denotes the feature map from the final stage of the visual encoder. The projection operator $\text{Proj}(\cdot)$ adjusts the spatial resolution of $\mathbf{S}^*$ to match that of $\hat{\mathbf{S}}_i$, and $\text{Conv}(\cdot)$ further refines the projected features. The resulting enhanced feature map $\mathbf{S}_{emd}$ incorporates both local and global semantic information, improving the ability of the model to reason about context-aware anomalies.

#### 3.5.2 PATCH-WISE ENHANCED MODULE

We introduce a Patch-wise Enhanced Module (PEM) with a Multi-Layer Perceptron (MLP) network. This module is designed to enhance the representational capacity of each patch through MLP during the training phase. In addition, as argued in (Gu et al., 2024), PEM helps maintain semantic consistency between the anomaly map and the LLM. The PEM process can be formally described as follows:

$$\mathbf{p}_i = \text{Flatten}(x[:, :, h_i : h_i + P, w_i : w_i + P]) \in \mathbb{R}^{B \times (P^2 \cdot C)}, \tag{8}$$

$$\mathbf{P} = \text{MLP}(\text{Norm}([\mathbf{p}_i]_{i=1}^N)) \in \mathbb{R}^{B \times N \times D}, \tag{9}$$

$$\mathbf{P}_k^{\text{reduced}} = \frac{1}{|G_k|} \sum_{i \in G_k} \mathbf{p}_i, \quad k = 1, \dots, K \tag{10}$$

$$\mathbf{P}^{\text{final}} = [\mathbf{P}_1^{\text{reduced}}, \dots, \mathbf{P}_K^{\text{reduced}}] \in \mathbb{R}^{B \times K \times D}, \tag{11}$$

where $x$ is the input anomaly map, $P$ denotes the patch size, $C$ is the number of channels, $B$ is the number of batch size, $N$ represents the number of spatial locations in the feature map, and $G_k$

represents the set of patch indices assigned to the $k$-th group. To obtain a compact and semantically rich representation from the anomaly map $x \in \mathbb{R}^{B \times C \times H \times W}$, the map is divided into patches of size $P \times P$, each of which is flattened into a vector $\mathbf{p}_i$. These vectors are then normalized and passed through an MLP to produce patch-level embeddings $\mathbf{P}$. To reduce computational overhead, the embeddings are grouped into $K$ clusters, and a mean representation is computed for each group. The final representation $\mathbf{P}^{\text{final}}$ is obtained by concatenating these group-wise averaged features.

---

**Algorithm 1** Self-Improvement Mechanism for Anomaly Detection

---

**Require:** Image $I$, Visual Encoder $V$, Text Encoder $T$, LLM $L$
**Ensure:** Final Response $R'$, Anomaly Score $S'$, Anomaly Map $M'$
 1: $V_{\text{feat}} \leftarrow V(I)$
 2: $P \leftarrow \text{InitialTextPrompt}()$
 3: $T_{\text{feat}} \leftarrow T(P)$
 4: $Z \leftarrow \text{Fuse}(V_{\text{feat}}, T_{\text{feat}})$
 5: $M \leftarrow \text{PredictAnomalyMap}(Z)$
 6: $S \leftarrow \text{ComputeScore}(M)$
 7: $R \leftarrow L(M)$
 8: $T_{\text{res}} \leftarrow T(R)$
 9: $Z' \leftarrow \text{Fuse}(V_{\text{feat}}, T_{\text{res}})$
10: $M' \leftarrow \text{PredictAnomalyMap}(Z')$
11: $S' \leftarrow \text{ComputeScore}(M')$
12: $R' \leftarrow L(M')$
13: **Return** $R', S', M'$

---

## 4 EXPERIMENTS

### 4.1 EXPERIMENTAL SETUPS

To evaluate the proposed model under the UZAD setting, we conducted experiments on nine benchmark datasets. These included MVTecAD (Bergmann et al., 2021), Visa (Zou et al., 2022), MPDD (Jezek et al., 2021), DTD (Aota et al., 2023), SDD (Tabernik et al., 2020), and BTAD (Mishra et al., 2021) for the industrial domain, as well as BrainMRI (Kanade & Gumaste, 2015), HeadCT (Kitamura, 2018), and Br35H (Hamada, 2020) for the medical domain. All input images were resized to $224 \times 224$ pixel resolution before being fed into the model. As the primary evaluation metric, we adopted the Area Under the Receiver Operating Characteristic (AUROC), which is widely used in anomaly detection to assess model performance. For implementation, we used ImageBind-Huge (Girdhar et al., 2023) as the image encoder and Vicuna-7B (Chiang et al., 2023) as the inference LLM. A linear projection layer was employed to bridge the modalities. The model was initialized with the pretrained weights from PandaGPT (Su et al., 2023). Training was conducted for one epoch using a single NVIDIA RTX 4090 GPU, with a batch size of 8 and a learning rate of 0.001. We employed the AdamW optimizer. We trained the model on MVTecAD and evaluated it on other industrial datasets, using weights pretrained on Visa for the MVTecAD evaluation. For the medical domain, we trained on Br35H and evaluated on other medical datasets, with the BrainMRI trained weights used to evaluate Br35H.

### 4.2 MAIN RESULTS

This section presents a quantitative evaluation of the proposed SIAD-LLM, comparing its performance with representative existing anomaly detection methods. A total of nine benchmark datasets were used in the experiments, spanning industrial and medical domains. The evaluation metrics include image-level AUROC, which evaluates anomaly detection performance at the image-level, and pixel-level AUROC, which evaluates the accuracy of localizing anomalous regions at the pixel-level. Pixel-level results on the medical datasets (BrainMRI, Br35H, HeadCT) are not reported, since ground-truth segmentation masks are not available for these datasets.

Table 1 shows the image and pixel-level AUROC results, where SIAD-LLM consistently outperforms existing methods across most datasets. Pixel-level AUROC results, in which SIAD-LLM

Table 1: Comparison of UZAD methods with image and pixel-level AUROC metric. Bold and underlining indicate best results and second-best results, respectively.

| Task | Method | Industrial domain | | | | | | Medical domain | | | Average |
|------|--------|---------|------|------|------|------|------|----------|--------|-------|---------|
| | | MVTecAD | Visa | MPDD | DTD | SDD | BTAD | BrainMRI | HeadCT | Br35H | |
| Image-level | CLIP | 60.9 | 49.1 | 44.9 | 75.2 | 40.1 | 59.3 | 91.0 | 56.5 | 80.2 | 61.9 |
| | AdaCLIP | 42.1 | 44.2 | 56.8 | 36.6 | 27.4 | 33.8 | 75.0 | 38.8 | 37.7 | 43.6 |
| | AnomalyCLIP | 52.0 | 39.4 | 49.3 | 59.7 | 30.9 | 68.3 | 76.1 | **81.5** | 59.4 | 57.4 |
| | AnomalyGPT | 71.1 | **63.1** | 59.6 | 88.9 | 55.5 | 70.4 | 90.1 | 58.2 | **87.0** | 71.5 |
| | FiLo | 40.2 | 48.5 | 53.2 | 45.7 | 19.8 | 67.7 | 74.3 | 68.3 | 25.2 | 49.2 |
| | AA-CLIP | 44.5 | 56.7 | 52.3 | 66.0 | 52.4 | 45.0 | 75.7 | 67.1 | 86.1 | 60.6 |
| | Ours | **72.8** | 62.2 | **66.9** | **93.7** | **62.6** | **73.1** | **93.9** | 76.0 | 83.1 | **76.0** |
| Pixel-level | CLIP | 53.4 | 51.3 | 63.1 | 28.3 | 12.3 | 47.5 | - | - | - | 42.7 |
| | AdaCLIP | 53.6 | 49.9 | 51.1 | 51.2 | 53.6 | 59.4 | - | - | - | 53.1 |
| | AnomalyCLIP | 56.7 | 66.7 | 48.7 | 77.6 | 77.6 | 51.3 | - | - | - | 63.1 |
| | AnomalyGPT | 84.8 | 85.9 | 90.3 | 94.4 | 81.0 | 80.2 | - | - | - | 86.1 |
| | FiLo | 53.1 | 60.0 | 60.1 | 72.1 | 55.4 | 59.9 | - | - | - | 60.1 |
| | AA-CLIP | 84.3 | 87.7 | 92.0 | 83.3 | **96.0** | 81.5 | - | - | - | 87.5 |
| | Ours | **86.0** | **88.1** | **93.2** | **97.2** | 87.5 | **88.2** | - | - | - | **90.0** |

Table 2: Comparison of UZAD methods with pseudo anomaly using image and pixel-level AUROC.

| Task | Method | Industrial domain | | | | | | Medical domain | | | Average |
|------|--------|---------|------|------|------|------|------|----------|--------|-------|---------|
| | | MVTecAD | Visa | MPDD | DTD | SDD | BTAD | BrainMRI | HeadCT | Br35H | |
| Image-level | AdaCLIP | 79.8 | 71.5 | 65.4 | 96.1 | 87.2 | 81.3 | 85.4 | 91.0 | 90.7 | 83.2 |
| | AnomalyCLIP | 79.2 | 69.1 | 76.0 | 96.3 | 97.5 | 77.5 | 97.9 | 99.1 | 94.8 | 87.5 |
| | FiLo | 73.3 | 65.2 | 56.9 | 95.8 | 81.0 | 85.1 | 78.9 | 52.9 | 69.9 | 76.2 |
| | AA-CLIP | 69.6 | 49.1 | 42.9 | 66.1 | 59.0 | 44.2 | 76.3 | 59.5 | 94.7 | 62.4 |
| | Ours | 72.8 | 62.2 | 66.9 | 93.7 | 62.6 | 73.1 | 93.9 | 76.0 | 83.1 | 76.0 |
| Pixel-level | AdaCLIP | 67.2 | 48.0 | 44.0 | 73.9 | 55.7 | 62.1 | - | - | - | 58.5 |
| | AnomalyCLIP | 87.6 | 93.5 | 94.0 | 97.8 | 98.1 | 93.6 | - | - | - | 94.1 |
| | FiLo | 83.7 | 89.2 | 85.2 | 95.9 | 97.3 | 85.8 | - | - | - | 89.5 |
| | AA-CLIP | 79.6 | 73.1 | 83.7 | 67.6 | 89.6 | 74.6 | - | - | - | 78.0 |
| | Ours | 86.0 | 88.1 | 93.2 | 97.2 | 87.5 | 88.2 | - | - | - | 90.0 |

achieves the highest localization performance on most datasets, with particularly notable improvements observed on datasets characterized by complex structures or high visual diversity. These results highlight the limitations of ZAD-based approaches, which suffer from reduced generalizability in the UZAD setting due to the absence of anomaly data during training. In contrast, the proposed SIAD-LLM demonstrates strong performance in both anomaly detection and localization, even without any access to anomalous samples, indicating its high generalizability across diverse domains and data distributions.

While the existing ZAD method learns both normal and abnormal boundaries, UZAD is designed to infer abnormality solely from the concept of normality. To investigate the impact of explicitly specifying abnormality, we conducted experiments to facilitate abnormality inference by providing pseudo anomalies to ZAD-based approaches, as shown in Table 2. In this process, AnomalyCLIP achieved high performance, whereas AdaCLIP and AA-CLIP, despite the explicit specifying of abnormality, still failed to sufficiently learn abnormality. These results demonstrate that the proposed UZAD task is considerably more challenging than ZAD and independent research value.

As shown in Fig. 4, we compared the qualitative results of our proposed model with existing methods. It is evident that CLIP (Radford et al., 2021), AnomalyCLIP (Zhou et al., 2023), and AdaCLIP (Cao et al., 2024) fail to accurately detect anomalies in the UZAD setting. In particular, CLIP tends to highlight only class-specific semantic regions of the object, revealing the limitations of directly applying a foundation model without task-specific finetuning. Although AA-CLIP (Ma et al., 2025) shows improved performance in detecting defect regions, it still attends to irrelevant class semantics and introduces considerable background noise, likely due to the absence of explicit anomaly supervision. In contrast, our SIAD-LLM effectively localizes defective regions while exhibiting strong robustness against background noise, demonstrating its superiority in both precision and contextual awareness.

Table 3: Ablation experiments of enhanced text prompt and module selection on industrial domain dataset. Results are reported using image and pixel-level AUROC.

**(a) Ablation experiments of SI (Self-improvement) and SP (Stage Prompt)**

| Method | MVTecAD | VisA | MPDD | DTD | SDD | BTAD | Average |
|---|---|---|---|---|---|---|---|
| w/o SI | (70.9, 84.8) | (60.9, 84.8) | (64.0, **93.2**) | (93.2, **97.2**) | (**65.1**, 78.2) | (61.0, 77.2) | (69.2, 85.9) |
| w/o SP | (66.6, 80.2) | (58.7, 87.8) | (65.1, 91.1) | (92.7, 96.1) | (62.0, **88.2**) | (71.9, 82.9) | (69.5, 87.7) |
| Full Model | (**72.8**, **86.0**) | (**62.2**, **88.1**) | (**66.9**, **93.2**) | (**93.7**, **97.2**) | (62.6, 87.5) | (**73.1**, **88.2**) | (**71.9**, **90.0**) |

**(b) Ablation experiments of EM and Adapter**

| Method | MVTecAD | VisA | MPDD | DTD | SDD | BTAD | Average |
|---|---|---|---|---|---|---|---|
| w/o EM | (68.7, 84.0) | (62.8, 87.1) | (62.9, 91.0) | (93.9, 96.9) | (59.2, 85.1) | (70.0, 82.1) | (69.6, 87.7) |
| w/o SEM | (71.4, 85.9) | (65.8, 88.3) | (62.8, 92.8) | (94.3, 97.3) | (61.0, 87.2) | (73.3, 88.0) | (71.4, 89.9) |
| w/o PEM | (69.8, 86.2) | (62.1, 87.3) | (65.6, 92.9) | (92.8, 96.8) | (**65.0**, 83.4) | (**74.0**, 81.7) | (71.6, 88.1) |
| w/o Adapter | (70.4, 82.7) | (**68.9**, **88.3**) | (58.4, 92.6) | (**95.6**, **97.9**) | (57.9, **89.7**) | (70.6, 82.6) | (70.3, 89.0) |
| Full Model | (**72.8**, **86.0**) | (62.2, 88.1) | (**66.9**, **93.2**) | (93.7, 97.2) | (62.6, 87.5) | (73.1, **88.2**) | (**71.9**, **90.0**) |

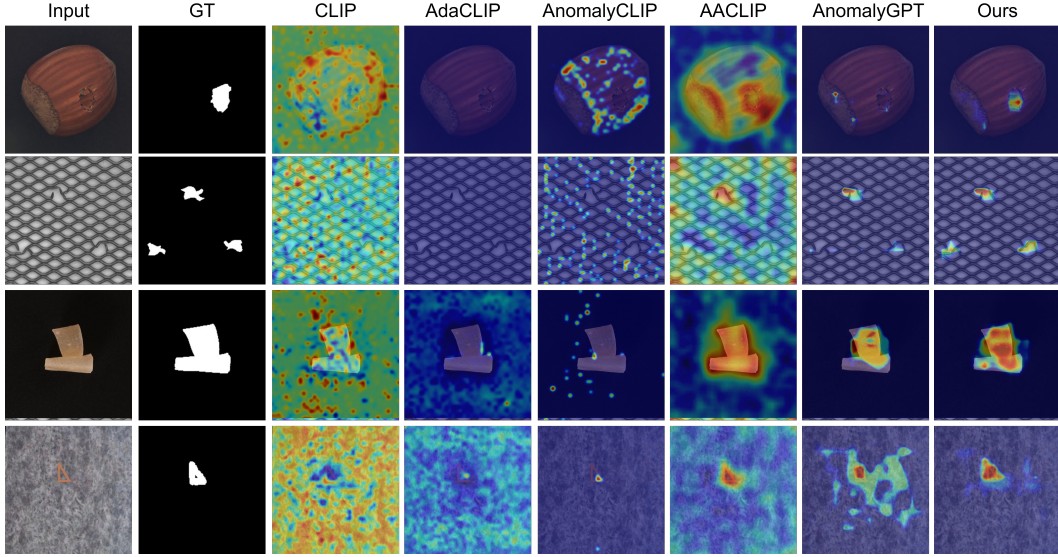

Figure 4: Qualitative results for anomaly localization on various domain datasets. From left to right: anomalous sample, ground-truth, predicted anomaly maps from other models, and our predicted anomaly map.

### 4.2.1 TEXT PROMPTS ENHANCEMENTS

As shown in Table 3 (a), we validate the effectiveness of structurally enhancing the expression and utilization of textual information on anomaly detection performance through ablation studies conducted on industrial-domain datasets. The results show that removing the self-improvement mechanism significantly degrades both image and pixel-level AUROC, indicating that using LLM-generated responses as text prompts is critical for learning normality and abnormality. Furthermore, excluding the stage prompt template notably lowers image-level AUROC, suggesting that introducing stage-wise variation in textual-visual interactions promotes diversity in the perceived scale of

Table 4: Ablation study under different adaptor coefficient($\lambda$) Values.

| $\lambda$ | MVTecAD | VisA | MPDD | DTD | SDD | BTAD | Average |
|---|---|---|---|---|---|---|---|
| 0.1 | **(72.8, 86.0)** | **(62.2, 88.1)** | **(66.9, 93.2)** | **(93.7, 97.2)** | (62.6, **87.5**) | **(73.1, 88.2)** | **(72.8, 86.0)** |
| 0.3 | (56.8, 63.4) | (53.7, 68.1) | (53.5, 67.6) | (72.2, 66.7) | (**71.3**, 68.5) | (55.2, 56.5) | (60.5, 65.1) |
| 0.5 | (49.6, 68.5) | (56.2, 62.8) | (62.6, 86.5) | (59.8, 78.0) | (71.2, 68.5) | (52.7, 64.8) | (58.7, 71.5) |

information within the image. These results demonstrate that not relying on predefined templates can support both representation and learning of normality and abnormality.

### 4.2.2 ENCODER ARCHITECTURE ENHANCEMENTS

As shown in Table 3 (b), we validate the effectiveness of encoder architecture enhancements through ablation studies on industrial datasets. Removing the EM leads to clear performance drops, indicating that strengthening visual representations is essential for accurate anomaly detection. The additional ablations on PEM and SEM further show that both patch-level and stage-level enhancements contribute individually to performance, confirming that multi-scale refinement of encoder features plays a crucial role in improving anomaly sensitivity. Moreover, the performance degradation observed when the adapter is removed highlights the necessity of finetuning the pretrained encoder for the anomaly detection task. Taken together, these findings demonstrate that finetuning the encoder and enhancing feature representations at multiple levels are both essential for achieving robust anomaly detection performance.

### 4.2.3 EFFECT OF ADAPTOR COEFFICIENT

The influence of the adaptor coefficient is examined through an ablation study summarized in Table 4. The adaptor controls the balance between the pretrained encoder features and the updated features produced through self-improvement, thereby regulating how strongly the text-driven signal alters the visual pathway. The results show that increasing the coefficient consistently reduces both image-level and pixel-level AUROC, indicating that amplifying the LLM-derived updates distorts the encoder representation and weakens its ability to capture normality. In contrast, the configuration with set to 0.1, which is the value used in all main experiments, preserves the stability of the pretrained encoder while allowing controlled refinement through the adaptor. These observations confirm that the coefficient functions as a structural stabilizer rather than a performance-tuning hyperparameter, and that maintaining a small value is essential for reliable anomaly detection.

## 5 CONCLUSION

This paper introduced SIAD-LLM, a novel framework for UZAD that requires no anomalous samples during training. By incorporating LLM-based self-improvement and stage prompt templates, the proposed method improved detection and localization. Extensive experiments on nine benchmarks across industrial and medical domains shows that SIAD-LLM consistently outperforms existing methods in both image and pixel-level anomaly detection, demonstrating strong generalization in anomaly-free settings. Although SIAD-LLM achieves strong zero-shot anomaly detection, the LLM backbone incurs high computational and time costs during both training and inference. As future work, we plan to explore lightweight or distilled LLM variants to reduce latency and memory usage while preserving detection accuracy.

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

# SUPPLEMENTARY MATERIAL

## A  USE OF LARGE LANGUAGE MODELS

We used large language models (LLMs) exclusively for grammar correction and minor typographical editing of the manuscript.

## B  LOSS FORMULATION AND SCORING DETAILS

We provide the full formulation of the loss used to optimize the model during training. The total loss combines focal, dice, and cross-entropy terms as follows:

$$\mathcal{L}_{\text{total}} = \alpha \cdot \mathcal{L}_{\text{focal}} + \beta \cdot \mathcal{L}_{\text{dice}} + \gamma \cdot \mathcal{L}_{\text{ce}}, \tag{12}$$

where $\alpha$, $\beta$, and $\gamma$ denote weighting coefficients for each term. In our implementation, we set all weights to 1. For inference, bilinear interpolation is used to resize the predicted anomaly map to the original image resolution. The maximum value of the map is then used as the image-level anomaly score. This formulation complements the high-resolution pixel-wise comparison performed during evaluation.

## C  STAGE PROMPTS DESIGN

To investigate the effect of stage prompt design, we conducted ablation experiments on the MVTecAD dataset by exploring different stage-wise prompt configurations. Table 5 presents the results of three stage prompt configurations. Case 1 represents a transition from low-level to high-level semantics, Case 2 represents a transition from

Table 5: Ablation results for different stage prompt configurations, reported by image and pixel-level AUROC.

| Setting | MVTecAD |
|---------|---------|
| Case 1 | (68.2, 84.9) |
| Case 2 | (69.2, 83.6) |
| Case 3 | (**72.8**, **86.0**) |

Table 6: **Stage prompt details.** This table shows the stage-specific prompts used in the results of Table 5. Case 3 represents the default setting used in our model.

| stage | Case1 | Case2 | Case3 |
|-------|-------|-------|-------|
| 1 | "very small region of" | "low level feature of" | "local feature of" |
| 2 | "small region of" | "mid level feature of" | "regional feature of" |
| 3 | "large region of" | "mid level feature of" | "regional feature of" |
| 4 | "very large region of" | "high level feature of" | "global feature of" |

small to large spatial focus, Case3 represents a transition from local to global features. The results show that Case 3 achieves the best performance, suggesting that aligning prompt semantics to the hierarchy of visual features facilitates more effective representation learning.

## D    TEXT PROMPT SETTING

Following conventional zero-shot anomaly detection setting, we utilize the compositional prompt ensemble to obtain initial prompts. Specifically, we consider state-level and template level. The complete text can be composed by replacing the token [c] in a template-level text with one of state-level text and replacing the token [o] with the object's name.

(a) *State*-level (normal)

- c := "[o]"
- c := "flawless [o]"
- c := "perfect [o]"
- c := "unblemished [o]"
- c := "[o] without flaw"
- c := "[o] without defect"
- c := "[o] without damage"

(b) *State*-level (anomaly)

- c := "damaged [o]"
- c := "broken [o]"
- c := "[o] with flaw"
- c := "[o] with defect"
- c := "[o] with damage"

(c) *Template*-level

• "a photo of a [c]."
• "a photo of the [c]."

Figure 5: Lists of state and template level prompts employed in this paper to construct text features.

## E    COMPARSION WITH TRAINING-FREE METHOD

To highlight the semantic grounding ability, we compared our model with training-free method AnoVL. Training-free methods have key limitations. Manual prompt engineering rely on hundreds of handcrafted prompts based on class names and defect types. This limits their scalability and applicability across diverse or unseen domains. Furthermore, semantic grounding is static and based on global features, making them less sensitive to small or localized anomalies critical in real-world tasks. Notably, on complex datasets like MPDD and BTAD, AnoVL shows a significant performance drop. In contrast, SIAD-LLM dynamically generates context-aware text via image-grounded QA and integrates it into learning. This allows strong generalization without manual tuning.

## F    DETAILD IMAGE DESCRIPTION

To facilitate anomaly reasoning, each prompt includes a concise textual description of the image content. This description outlines the object class and its expected properties under normal conditions, serving as contextual grounding for the LLM. During training, prompts are constructed in the following format:

```
 Human:   E_img </Img> E_prompt [Image Description] Is there
any anomaly in the image?    Assistant:
```

Table 7: Comparison of with training-free methods with pixel-level AUROC metric.

| Method | Industrial domain | | | | | | Medical domain | | | Average |
|--------|---------|------|------|------|------|------|----------|--------|-------|---------|
| | MVTecAD | Visa | MPDD | DTD | SDD | BTAD | BrainMRI | HeadCT | Br35H | |
| AnoVL | 85.7 | 85.8 | 59.9 | 93.2 | 95.2 | 75.2 | - | - | - | 82.5 |
| Ours | 86.0 | 88.1 | 93.2 | 97.2 | 87.5 | 88.2 | - | - | - | 90.0 |

The descriptions provided for each category in the various datasets are summarized in the table below and are used to guide the LLM in understanding what defines a normal instance.

Table 8: Detailed image description for every category in MVTecAD dataset. The description is used to construct prompts for anomaly detection.

| Class | Image Description |
|-------|-------------------|
| bottle | This is a photo of a bottle for anomaly detection, which should be round, without any damage, flaw, defect, scratch, hole or broken part. |
| cable | This is a photo of three cables for anomaly detection, cables cannot be missed or swapped, which should be without any damage, flaw, defect, scratch, hole or broken part. |
| capsule | This is a photo of a capsule for anomaly detection, which should be black and orange, with print '500', without any damage, flaw, defect, scratch, hole or broken part. |
| carpet | This is a photo of carpet for anomaly detection, which should be without any damage, flaw, defect, scratch, hole or broken part. |
| grid | This is a photo of grid for anomaly detection, which should be without any damage, flaw, defect, scratch, hole or broken part. |
| hazelnut | This is a photo of a hazelnut for anomaly detection, which should be without any damage, flaw, defect, scratch, hole or broken part. |
| leather | This is a photo of leather for anomaly detection, which should be brown and without any damage, flaw, defect, scratch, hole or broken part. |
| metal_nut | This is a photo of a metal nut for anomaly detection, which should be without any damage, flaw, defect, scratch, hole or broken part, and shouldn't be fliped. |
| pill | This is a photo of a pill for anomaly detection, which should be white, with print 'FF' and red patterns, without any damage, flaw, defect, scratch, hole or broken part. |
| screw | This is a photo of a screw for anomaly detection, which tail should be sharp, and without any damage, flaw, defect, scratch, hole or broken part. |
| tile | This is a photo of tile for anomaly detection, which should be without any damage, flaw, defect, scratch, hole or broken part. |
| toothbrush | This is a photo of a toothbrush for anomaly detection, which should be without any damage, flaw, defect, scratch, hole or broken part. |
| transistor | This is a photo of a transistor for anomaly detection, which should be without any damage, flaw, defect, scratch, hole or broken part. |
| wood | This is a photo of wood for anomaly detection, which should be brown with patterns, without any damage, flaw, defect, scratch, hole or broken part. |
| zipper | This is a photo of a zipper for anomaly detection, which should be without any damage, flaw, defect, scratch, hole or broken part. |

Table 9: Detailed image description for every category in Visa dataset. The description is used to construct prompts for anomaly detection.

| Class | Image Description |
|---|---|
| candle | This is a photo of 4 candles for anomaly detection, every candle should be round, without any damage, flaw, defect, scratch, hole or broken part. |
| capsules | This is a photo of many small capsules for anomaly detection, every capsule is green, should be without any damage, flaw, defect, scratch, hole or broken part. |
| cashew | This is a photo of a cashew for anomaly detection, which should be without any damage, flaw, defect, scratch, hole or broken part. |
| chewinggum | This is a photo of a chewinggom for anomaly detection, which should be white, without any damage, flaw, defect, scratch, hole or broken part. |
| fryum | This is a photo of a fryum for anomaly detection on green background, which should be without any damage, flaw, defect, scratch, hole or broken part. |
| macaroni1 | This is a photo of 4 macaronis for anomaly detection, which should be without any damage, flaw, defect, scratch, hole or broken part. |
| macaroni2 | This is a photo of 4 macaronis for anomaly detection, which should be without any damage, flaw, defect, scratch, hole or broken part. |
| pcb1 | This is a photo of pcb for anomaly detection, which should be without any damage, flaw, defect, scratch, hole or broken part. |
| pcb2 | This is a photo of pcb for anomaly detection, which should be without any damage, flaw, defect, scratch, hole or broken part. |
| pcb3 | This is a photo of pcb for anomaly detection, which should be without any damage, flaw, defect, scratch, hole or broken part. |
| pcb4 | This is a photo of pcb for anomaly detection, which should be without any damage, flaw, defect, scratch, hole or broken part. |
| pipe_fryum | This is a photo of a pipe fryum for anomaly detection, which should be without any damage, flaw, defect, scratch, hole or broken part. |

Table 10: Detailed image description for every category in MPDD dataset. The description is used to construct prompts for anomaly detection.

| Class | Image Description |
|---|---|
| bracket_black | This is a photo of a bracket_black for anomaly detection, which should be black and without any damage, flaw, defect, scratch, hole or broken part. |
| bracket_brown | This is a photo of a bracket_brown for anomaly detection, which should be brown and without any damage, flaw, defect, scratch, hole or broken part. |
| bracket_white | This is a photo of a bracket_white for anomaly detection, which should be white and without any damage, flaw, defect, scratch, hole or broken part. |
| connector | This is a photo of a connector for anomaly detection, which should be without any damage, flaw, defect, scratch, hole or broken part. |
| metal_plate | This is a photo of a metal_plate for anomaly detection, which should be without any damage, rust, flaw, defect, scratch, hole or broken part. |
| tubes | This is a photo of a tubes for anomaly detection, which should be without any damage, flaw, defect, scratch, hole or broken part. |

Table 11: Detailed image description for every category in DTD dataset. The description is used to construct prompts for anomaly detection.

| Class | Image Description |
| --- | --- |
| Woven_001 | This is a photo of a Woven_001 for anomaly detection, which should be without any damage, flaw, defect, scratch, hole or broken part. |
| Woven_127 | This is a photo of a Woven_127 for anomaly detection, which should be without any damage, flaw, defect, scratch, hole or broken part. |
| Stratified_154 | This is a photo of a Stratified_154 for anomaly detection, which should be without any damage, flaw, defect, scratch, hole or broken part. |
| Blotchy_099 | This is a photo of a Blotchy_099 for anomaly detection, which should be without any damage, flaw, defect, scratch, hole or broken part. |
| Woven_068 | This is a photo of a Woven_068 for anomaly detection, which should be without any damage, flaw, defect, scratch, hole or broken part. |
| Woven_125 | This is a photo of a Woven_125 for anomaly detection, which should be without any damage, flaw, defect, scratch, hole or broken part. |
| Marbled_078 | This is a photo of a Marbled_078 for anomaly detection, which should be without any damage, flaw, defect, scratch, hole or broken part. |
| Perforated_037 | This is a photo of a Perforated_037 for anomaly detection, which should be without any damage, flaw, defect, scratch, hole or broken part. |
| Mesh_114 | This is a photo of a Mesh_114 for anomaly detection, which should be without any damage, flaw, defect, scratch, hole or broken part. |
| Fibrous_183 | This is a photo of a Fibrous_183 for anomaly detection, which should be without any damage, flaw, defect, scratch, hole or broken part. |
| Matted_069 | This is a photo of a Matted_069 for anomaly detection, which should be without any damage, flaw, defect, scratch, hole or broken part. |
| Woven_104 | This is a photo of a Woven_104 for anomaly detection, which should be without any damage, flaw, defect, scratch, hole or broken part. |

Table 12: Detailed image description for every category in BTAD dataset. The description is used to construct prompts for anomaly detection.

| Class | Image Description |
| --- | --- |
| 01 | This is a photo of a 01 for anomaly detection, which should be round, without any damage, flaw, defect, scratch, hole or broken part. |
| 02 | This is a photo of a 02 for anomaly detection, which should be round, without any damage, flaw, defect, scratch, hole or broken part. |
| 03 | This is a photo of a 03 for anomaly detection, which should be round, without any damage, flaw, defect, scratch, hole or broken part. |

Table 13: Detailed image description for every category in SDD dataset. The description is used to construct prompts for anomaly detection.

| Class | Image Description |
| --- | --- |
| SDD | This is a photo of a electrical_commutators for anomaly detection, which should be without any damage, flaw, defect, scratch, hole or broken part. |

Table 14: Detailed image description for every category in Br35H dataset. The description is used to construct prompts for anomaly detection.

| Class | Image Description |
| --- | --- |
| brain | This is a photo of a brain for medical anomaly detection, which should be without any damage, flaw, defect, scratch, hole or broken part. |

Table 15: Detailed image description for every category in BrainMRI dataset. The description is used to construct prompts for anomaly detection.

| Class | Image Description |
| --- | --- |
| brain | This is a photo of a brain for medical anomaly detection, which should be without any damage, flaw, defect, scratch, hole or broken part. |

Table 16: Detailed image description for every category in HeadCT dataset. The description is used to construct prompts for anomaly detection.

| Class | Image Description |
| --- | --- |
| brain | This is a photo of a brain for medical anomaly detection, which should be without any damage, flaw, defect, scratch, hole or broken part. |

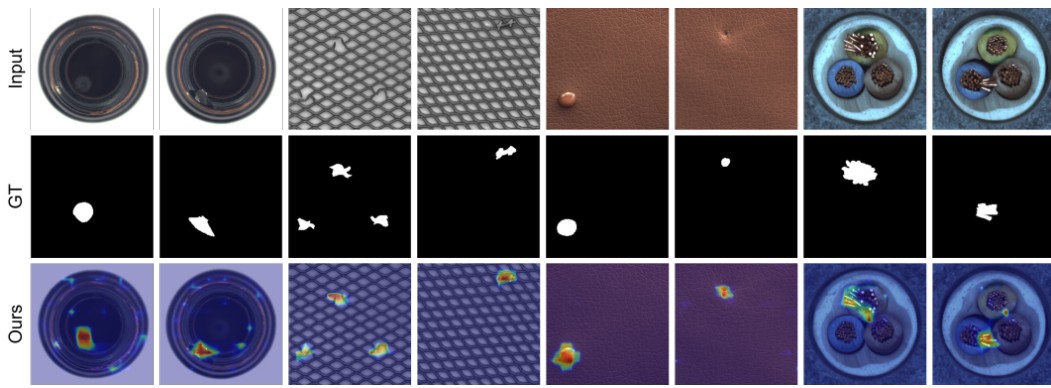

Figure 6: Qualitative results for anomaly localization on MVTecAD dataset. From top to bottom: anomalous sample, our predicted anomaly map.

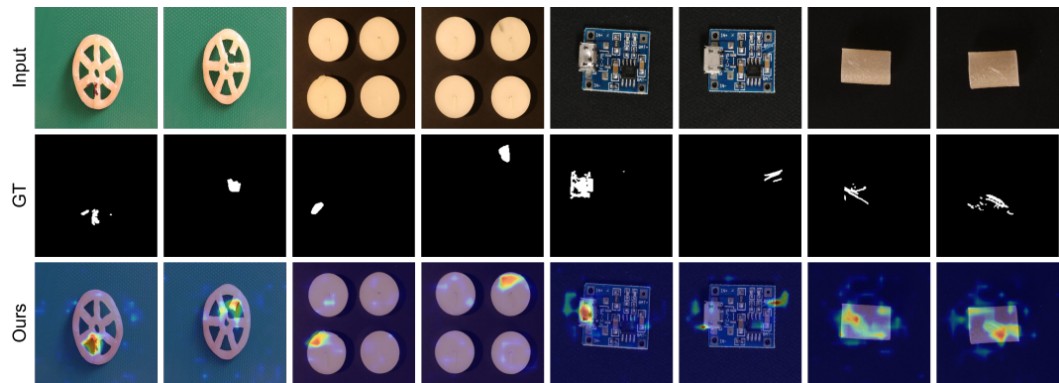

Figure 7: Qualitative results for anomaly localization on VisA dataset. From top to bottom: anomalous sample, our predicted anomaly map.

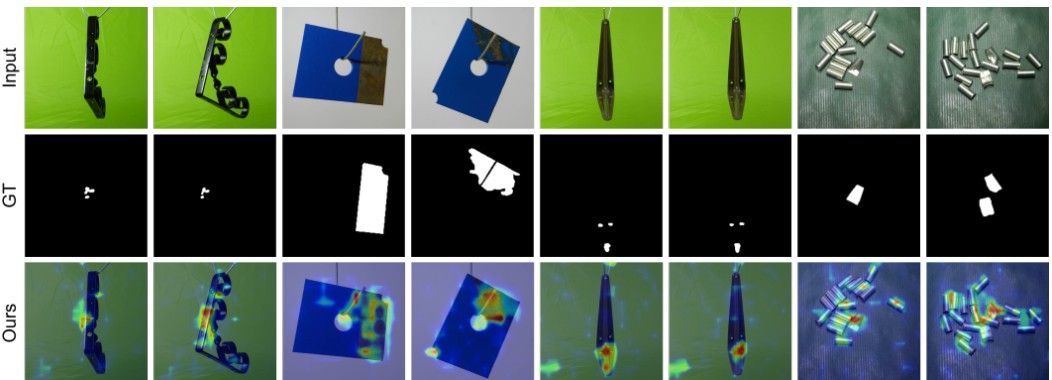

Figure 8: Qualitative results for anomaly localization on MPDD dataset. From top to bottom: anomalous sample, our predicted anomaly map.

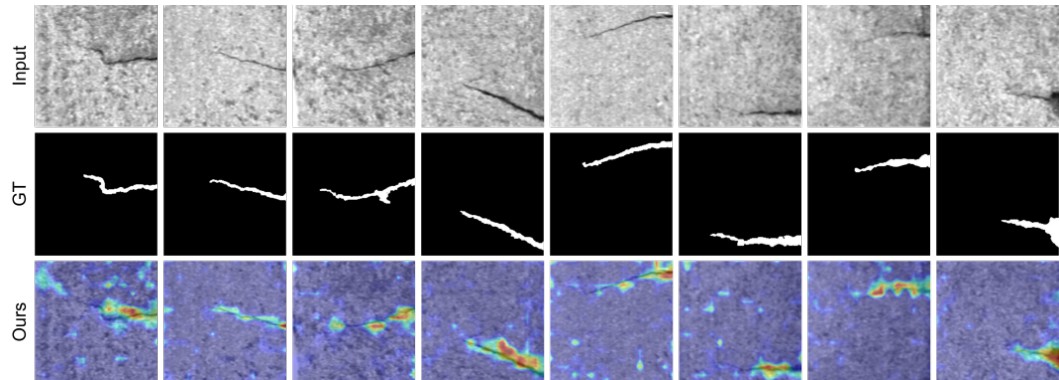

Figure 9: Qualitative results for anomaly localization on SDD dataset. From top to bottom: anomalous sample, our predicted anomaly map.

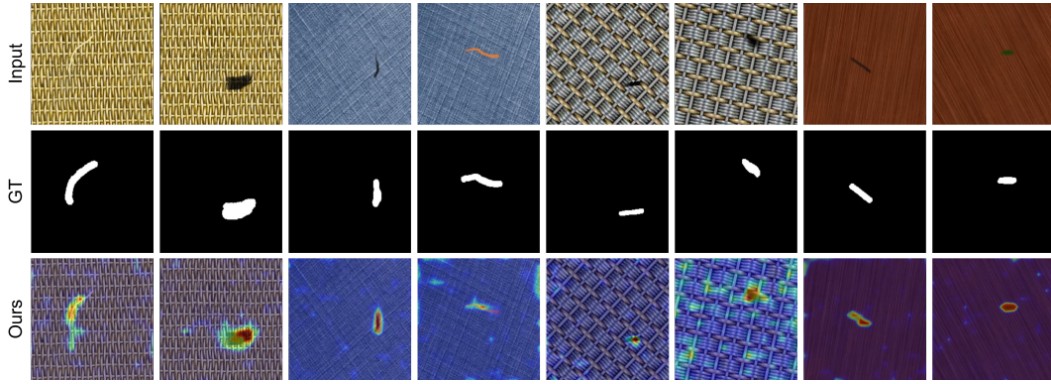

Figure 10: Qualitative results for anomaly localization on DTD dataset. From top to bottom: anomalous sample, our predicted anomaly map.

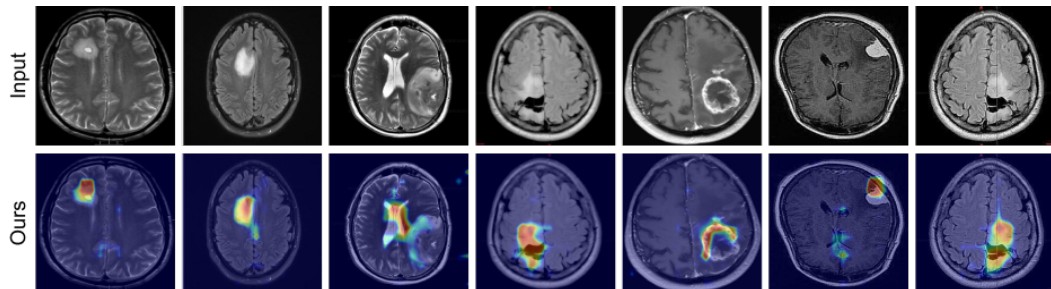

Figure 11: Qualitative results for anomaly localization on BrainMRI dataset. From top to bottom: anomalous sample, our predicted anomaly map. Note that BrainMRI does not provide ground-truth localization annotations.

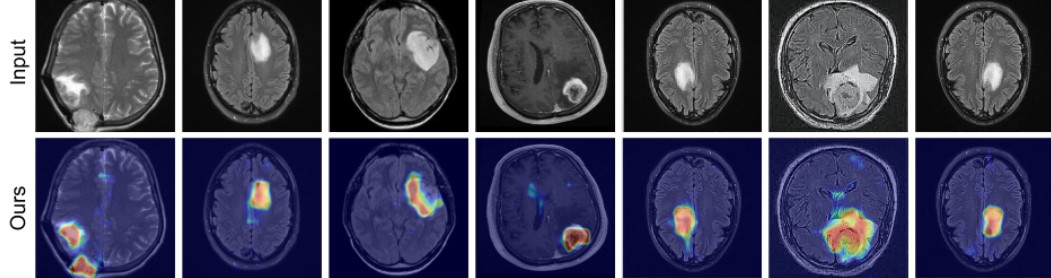

Figure 12: Qualitative results for anomaly localization on Br35H dataset. From top to bottom: anomalous sample, our predicted anomaly map. Note that Br35H does not provide ground-truth localization annotations.

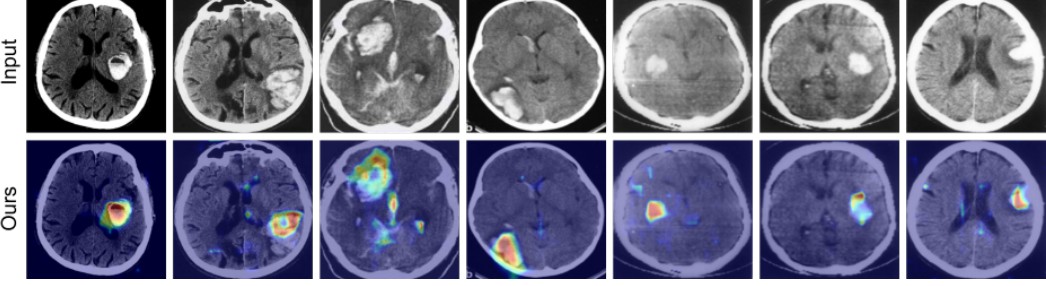

Figure 13: Qualitative results for anomaly localization on HeadCT dataset. From top to bottom: anomalous sample, our predicted anomaly map. Note that HeadCT does not provide ground-truth localization annotations.

