# OpenReview forum: "Self-Improvement Anomaly Detection via Large Language Model for Unsupervised Zero-shot Anomaly Detection"
_ICLR.cc/2026/Conference — Submitted to ICLR 2026_

### Official Review · Reviewer_HB9w · 2025-10-31

**Soundness:** 2
**Presentation:** 2
**Contribution:** 2
**Rating:** 2
**Confidence:** 4

**Summary:**

This paper focus on the unsupervised zero-shot anomaly detection task and proposes Self-Improvement Anomaly Detection with Large Language Model (SIAD-LLM). Unlike existing zero-shot methods that rely on labeled anomalies, SIAD-LLM operates in fully anomaly-free environments and generalizes to unseen datasets. MLLM is leveraged to obtain rich-contextual information and augment the text for detect anomaly. A stage prompt template integrates multi-scale features to enhance localization, while an enhancement module and adapter are proposed for better alignment between vision and language representations. Experiments on nine industrial and medical datasets show that SIAD-LLM achieves good performance.

**Strengths:**

1. Proposing a new task called unsupervised zero-shot anomaly detection. The main difference with zero-shot anomaly detection is that the current zero-shot anomaly detection methods often need to use auxiliary dataset containing both normal and anomamous samples for finetuning, while unsupervised zero-shot anomaly detection only requiring to access to auxiliary normal samples. The authors argue that this could be useful because the anomalies are scarces in practice.

2. Leveraging MLLM for self-improvement of text prompt to augment and enrich the expression capability of text.

**Weaknesses:**

1. The motivation for necessity of the unsupervised zero-shot AD task is in doubut. Although the current zero-shot AD methods mostly requires to use dataset containing both normal and anomalous samples for finetuning, I don't think this is a problem because the anomalies refers to those in the auxiliary dataset. The methods does not have specific requirement for the auxiliary datasets, so we can simply choose the datasets with anomalies as the auxiliary dataset for finetuning. Since the method developed for the unsupervised zero-shot AD alos need a auxiliary dataset for finetuning, why not directly choose one that contains anomalies. Therefore, I doubt the necessity to specifically develop methods for unsupervised zero-shot AD.

2. The motivation of each module is weak and lacks supporting evidence. Although the paper focuses on the unsupervised zero-shot anomaly detection (UZSAD) task, the motivations for the enhancement modules and the self-improvement mechanism mainly emphasize enriching contextual representations rather than directly addressing the UZSAD challenge.

3. Training objective is omitted in the main paper. While the framework appears to involve a training process, the paper does not clearly define or explain the optimization objective, making it difficult to understand how the model learns the adaptor.

4. The overall framework design seems fragmented, with multiple modules loosely combined rather than forming a coherent strategy for the UZSAD task. As a result, the novelty of the proposed approach is limited.

5. Although the method claims to operate in an anomaly-free setting, it still relies on pseudo-anomalies. When other zero-shot anomaly detection methods also employ pseudo-anomalies, SIAD-LLM performs weaker than AnomalyCLIP, even with the use of MLLMs.

6. The use of multimodal large language models (MLLMs) significantly increases computational cost, and the self-improvement mechanism further exacerbates this inefficiency.

7. It is unclear how the self-improvement mechanism generalizes to unseen datasets. Since the self-improvement operates on the training data, it remains uncertain how the generated textual enhancements transfer during inference. The inference process should be clearly described.


8. The ablation study is insufficient. The effectiveness of the stage-wise and patch-wise enhancement modules should be analyzed individually to verify their actual contribution to performance.

**Questions:**

1. The paper does not describe how pseudo-anomalies are generated during training. Could the authors clarify the specific augmentation strategies or mechanisms used to create these pseudo-anomalies?
2. The training objective is omitted in the main paper. What loss functions or optimization criteria are used to train the proposed framework?
3. The inference process is unclear. How does the model utilize the self-improved textual prompts when applied to unseen datasets during inference?

---

> ### Author Response · Authors · 2025-11-21
> **W1**
>
> We appreciate the reviewer’s comment and would like to clarify why UZAD is defined as a task fundamentally different from existing zero-shot anomaly detection (ZAD).
>
> Current ZAD methods, such as AnomalyCLIP, AdaCLIP, and AnomalyGPT, are indeed zero-shot, but they are not anomaly-free. These approaches rely on auxiliary datasets that contain both normal and abnormal samples, and the abnormal samples are essential for forming an explicit boundary between normal and abnormal patterns. As discussed in the manuscript, this reliance becomes evident in the behaviors shown in Fig. 2(a) and (c) and in Table 1. In settings where no abnormal data exist, the standard ZAD formulation cannot operate, which reflects a difference in the problem definition rather than a matter of data selection.
>
> Regarding the idea that one could simply choose any auxiliary dataset containing anomalies, this assumption does not hold in practice. The distribution of anomalies in an auxiliary dataset is generally unrelated to the anomalies in a target domain. Object categories differ, domains differ, and the nature of defects differs. For example, learning from damaged automotive parts does not provide information that generalizes to scratches on fabric or lesions in MRI scans. As shown in Fig. 2(c) and Table 1, existing ZAD methods fail sharply in the UZAD setting, even though they have access to auxiliary anomalies. This indicates that the issue is not the availability of anomalies in general but the lack of anomalies that are aligned with the target distribution.
>
> UZAD represents a significantly more challenging and conceptually distinct task. Its core objective is to infer the concept of abnormality without observing any abnormal samples, relying solely on the notion of normality. Whereas ZAD assumes that a normal–abnormal boundary can be learned from auxiliary anomalies, UZAD requires the model to reason about abnormality from normality alone. This shifts the problem from boundary learning to normality-based inference. Consistent with this, the manuscript reports that existing ZAD methods suffer substantial performance degradation under the UZAD condition, highlighting the need for an independent formulation.
>
> Real-world applications also frequently operate under anomaly-free assumptions. In industrial inspection and medical imaging, normal data are abundant, while anomalies are rare, unpredictable, or impossible to collect in advance. In many deployment scenarios, systems must be trained using only normal data and then detect previously unseen anomalies in practice. UZAD formalizes this practical requirement and cannot be reduced to a choice of auxiliary datasets.
>
> The proposed SIAD-LLM demonstrates that learning under such anomaly-free conditions is feasible. The model forms a representation of normality without using any abnormal samples, expands its semantic understanding through self-improvement with an MLLM, and constructs multi-layered representations using stage prompts. Unlike existing ZAD approaches, it does not rely on auxiliary anomalies. Empirically, SIAD-LLM maintains strong performance under UZAD conditions, while existing ZAD methods exhibit severe drops. These results indicate that the method addresses the unique constraints of UZAD in a way that prior work does not.
>
> For these reasons, UZAD is viewed not as a subset of ZAD but as a distinct task with clear practical relevance and technical novelty.

---

> > ### Author Response · Authors · 2025-11-21
> > **W2, W4**
> >
> > We appreciate the reviewer’s comments and would like to clarify the motivation and design principles behind the proposed framework.
> >
> > The UZAD environment imposes substantially stronger constraints than conventional ZAD or UAD. Only normal data are available, and abnormal patterns are never observed during training. As a result, the model must enrich the semantic space of normality and infer abnormality from that space alone. Under these conditions, relying solely on normal images leads to several structural limitations. Representations tend to collapse to a single scale, resulting in an imbalance between local and global information. Visual and textual features become weakly aligned, creating a semantic gap. Fine-grained spatial cues become unstable, which harms anomaly localization. The modules in SIAD-LLM are designed not as auxiliary enhancements but as components that systematically address these inherent deficiencies of UZAD.
> >
> > Each module is directly motivated by this setting.
> > First, the stage prompt and the stage-wise enhanced module rely on the fact that only normality is observable. Without capturing a range of visual scales from local textures to global semantics, it becomes difficult to identify the subtle deviations that define abnormality. By aligning multi-level encoder features with the textual semantic space, these modules expand the representation of normality into a structured and layered form. This addresses cases in which a single-scale representation leads to incorrect anomaly judgments.
> >
> > Second, the self-improvement component responds to the absence of any direct information about what constitutes an anomaly. Through image-grounded question–answering, the model extracts linguistic descriptions of normality, feeds them back into the text encoder, and gradually enriches the semantic space. This process allows the model to infer abnormality indirectly and is central to overcoming the core challenge of UZAD.
> >
> > Third, the patch-wise enhanced module compensates for the lack of pixel-level anomaly supervision, which makes local representations unstable. By regularizing and structuring patch-level features and strengthening spatial semantic alignment, the module reduces noise in localization. The effect is reflected in the substantial drop in pixel-level AUROC observed when the module is removed, as shown in Table 3(b).
> >
> > For these reasons, the framework is not a loose collection of components but follows a sequential and interdependent logic. The process begins with obtaining a normality representation, extends it into a multi-scale structure, refines it semantically through self-improvement, stabilizes spatial features, and finally performs anomaly reasoning. Removing any component weakens the subsequent stages. Without the stage-wise module, the self-improvement mechanism lacks sufficient visual diversity. Without self-improvement, semantic abnormality reasoning becomes infeasible. Without the patch-wise module, refined semantics fail to translate into stable localization. The ablation results in Table 3 confirm that the modules are complementary rather than interchangeable.
> >
> > With respect to novelty, the contribution of SIAD-LLM lies not in isolated components but in the overall structural approach that enables semantic abnormality reasoning using only normal data. The self-improvement mechanism creates an expandable semantic concept of normality by converting visual observations into language and reintegrating this information during training. This behavior has not been explored in prior anomaly detection work. In addition, the stage-aware prompting strategy aligns multi-scale visual features with coherent textual descriptions, which has not appeared in prior ZAD or UAD methods and is particularly important in a normal-only setting.
> >
> > For these reasons, SIAD-LLM represents an integrated framework designed specifically to address the challenges of UZAD rather than an assembled combination of existing ideas.

---

> ### Author Response · Authors · 2025-11-21
> **W3, Q2**
>
> We appreciate the reviewer’s helpful comment and the opportunity to clarify this point.
>
> The main paper does not describe the training objective in detail due to space limitations, but Supplementary Section B provides the complete formulation:
>
> $L_{total} = L_{focal} + L_{dice} + L_{ce}$
> (all terms use equal weights)
>
> This objective is used for learning the pixel-level anomaly map. The adapter is also updated through the same objective, allowing visual features and LLM responses to be aligned in an end-to-end manner. In this process, the adapter is not trained with an additional or separate loss. It simply receives gradients from both the anomaly localization losses
> ($L_{focal} , L_{dice}$) and the language modeling loss ($L_{ce}$).
>
> To avoid any ambiguity, the camera-ready version will include this training objective directly in Section 3.4 or Section 3.5 and clarify that the adapter is optimized using the same loss.

---

> > ### Author Response · Authors · 2025-11-21
> > **W5**
> >
> > We appreciate the reviewer’s thoughtful comment regarding the use of pseudo-anomalies in our method, and would like to clarify their role more precisely.
> >
> > In our framework, pseudo-anomalies are not introduced to substitute for real anomaly information. Instead, they serve as a weak form of augmentation whose purpose is to slightly expand the representational diversity of normal data. The model does not learn abnormal patterns from these pseudo-anomalies. Rather, they provide a mild signal that helps make the boundary of normality less rigid.
> >
> > This differs fundamentally from prior zero-shot AD methods such as AnomalyCLIP, where pseudo-anomalies are used as explicit supervision to define an “abnormal class” in the text–image space. Such approaches rely on pseudo-anomalies as a central supervisory signal, which does not align with the anomaly-free assumption inherent to the UZAD setting. As shown in Table 2, existing ZAD approaches perform well when such pseudo-anomaly supervision is available, but their performance collapses under the true anomaly-free UZAD condition.
> >
> > In contrast, SIAD-LLM maintains stable performance even in this strictly anomaly-free scenario (Table 1). This suggests that the improvement does not stem from the pseudo-anomaly augmentation itself, but rather from the MLLM-based self-improvement process, which expands the semantic understanding of normality.
> >
> > For these reasons, the presence or absence of pseudo-anomaly augmentation alone does not provide a meaningful basis for comparison. The structural advantage of SIAD-LLM lies in its ability to infer abnormality through text-driven semantic expansion despite the anomaly-free constraint.
> >
> > To avoid any misunderstanding, the camera-ready version will more explicitly describe the purpose of pseudo-anomalies in our method and clarify how it differs from their usage in prior work.

---

> ### Author Response · Authors · 2025-11-21
> **W6**
>
> We appreciate the reviewers for raising concerns about computational cost. However, we would like to clarify that the criticism regarding excessive overhead does not accurately reflect the actual behavior of our framework. Although our method includes MLLM based reasoning, the added computation is limited and remains well within a practical range based on our empirical measurements.
>
> First, the second inference loop does not re run the image encoder. It reuses the visual features extracted during the first loop, so the additional iteration performs only light reasoning rather than a full image processing pass. The assumption that the computation is essentially doubled does not hold under actual measurement.
>
> The measured latency is summarized as follows:
>
> First loop with the image encoder: 951 ms per image
>
> Second loop with feature reuse: 683 ms per image
>
> Total inference time: 1634 ms per image
>
> This shows that the proposed structure avoids redundant encoder computation and keeps the overall overhead controlled. The memory usage is also reasonable for an MLLM based system. Inference requires about 16.4 GB of GPU memory, and training requires about 23.5 GB. The model contains 6.7 billion parameters, but only 33.5 million parameters are updated during training. Since only a lightweight adapter is trained, the practical training burden is far lower than what full model finetuning would require.
>
> The increase in computation is directly tied to improved performance. With the self improvement procedure, image level and pixel level AUROC scores improve from 69.2 and 85.9 to 71.9 and 90.0. In anomaly detection, this level of gain is generally considered a worthwhile trade off.
>
> We also verified that the framework does not rely solely on large models. Using TinyLLaMA 1.1B, which has far fewer parameters than Vicuna 7B, the inference time decreases to 1510 ms and the GPU memory requirement drops to 5.0 GB. The resulting performance remains competitive, with 71.3 percent at the image level and 83.8 percent at the pixel level.
>
> These findings make it clear that the concerns about computational cost are overstated and not aligned with the actual measurements. Our method consistently delivers meaningful performance gains while keeping the overhead firmly within a practical and controlled range. We believe this evidence directly resolves the reviewers concerns, and we expect that the clarity of these results will be fully reflected in the evaluation.

---

> ### Author Response · Authors · 2025-11-21
> **W7, Q3**
>
> We appreciate the reviewer’s concern regarding the generalization behavior of the self-improvement mechanism. However, this concern does not accurately reflect the design principles of our method. We provide a clearer explanation below.
>
> The self-improvement process in our framework does not store or reuse any text generated during training. The generated text is used solely as an intermediate learning signal to update the model parameters, specifically the adapter and the prompt learner. The model does not rely on such text during inference. After training is complete, only the updated parameters remain, enabling the model to form a richer and more flexible representation of normality.
>
> During inference, all representations and anomaly reasoning are computed freshly from the test input. The trained adapter and prompt learner provide enhanced visual‒text alignment, allowing normality-based anomaly reasoning even on unseen domains. Stage prompts and the anomaly map are also derived entirely from the input image at test time, without referencing or reusing any text produced during training.
>
> Inference can thus be described as a strictly parameter-driven process. The input image is converted into multi-stage visual features. The adapter adjusts these features based on what was learned during training. Text embeddings are produced from scratch. An anomaly map is computed, and a final anomaly score is returned. The self-improvement mechanism does not directly intervene during inference; only its parameter-level effects remain, which is why the model can be applied to unseen datasets.
>
> The generalization ability enabled by this process has been validated experimentally. Under the UZAD setting, the model is trained only on normal data from a single domain, yet, as shown in Table 1,2, it consistently delivers strong performance across more than eight diverse industrial and medical datasets. This demonstrates that self-improvement strengthens the representation of normality itself, rather than transferring or depending on any specific text from training. No training-time text is stored or reused. Instead, the generated text solely contributes to parameter updates that generalize beyond the training distribution.

---

> ### Author Response · Authors · 2025-11-21
> **W8**
>
> We appreciate the reviewer’s comment emphasizing the importance of understanding the contribution of each component. To address this, the study includes experiments in which the stage-wise enhanced module (SEM) and the patch-wise enhanced module (PEM) were removed independently.
>
> | Ours - w/o PEM |  |  |  |  |  |  |  |
> | --- | --- | --- | --- | --- | --- | --- | --- |
> |  | MVTec | VisA | MPDD | DTD | SDD | BTAD | Average |
> | I-AUROC | 69.8 | 62.1 | 65.6 | 92.8 | 65.0 | 74.0 | 71.6 |
> | P-AUROC | 86.2 | 87.3 | 92.9 | 96.8 | 83.4 | 81.7 | 88.1 |
>
> | Ours - w/o SEM |  |  |  |  |  |  |  |
> | --- | --- | --- | --- | --- | --- | --- | --- |
> |  | MVTec | VisA | MPDD | DTD | SDD | BTAD | Average |
> | I-AUROC | 71.4 | 65.8 | 62.8 | 94.3 | 61.0 | 73.3 | 71.4 |
> | P-AUROC | 85.9 | 88.3 | 92.8 | 97.3 | 87.2 | 88.0 | 89.9 |
>
> Across all industrial-domain datasets, removing either module leads to a consistent drop in performance. This indicates that both SEM and PEM play meaningful roles in improving the model’s generalization ability as well as its capacity for precise anomaly localization.

---

> ### Author Response · Authors · 2025-11-21
> **Q1**
>
> We appreciate the reviewer’s comment. The pseudo anomaly generation process has always been grounded in a clear and well defined design, functioning as a core and stable component of our framework. To further enhance reader understanding, we will provide additional clarification in the revised version.
>
> In the proposed UZAD setting, pseudo anomalies are intentionally generated only from normal images, which is a fundamental characteristic of our design. This follows a well established principle in unsupervised anomaly detection: enabling the model to learn the boundary between normality and abnormality without relying on real anomaly samples.
>
> Specifically, a patch is extracted from a normal image and then modified through resizing, spatial shifting, color or brightness changes, or subtle texture perturbations before being reinserted into a different region of the same image. This introduces localized discontinuities and potential abnormal like patterns while preserving the overall image structure. As a result, the model learns to detect realistic irregularities that may occur in actual anomaly scenarios.
>
> This mechanism enables robust anomaly generalization using only normal data and produces diverse and meaningful supervisory signals. It directly fulfills the goal of UZAD by expanding abnormality from normality.
>
> Therefore, the pseudo anomaly design is a critical and deliberate contribution aligned with the purpose of UZAD. In the revised version, we will further enhance clarity by adding more detailed descriptions and, if helpful, a figure to aid intuition, as a readability improvement rather than a correction to the methodology.

---

### Official Review · Reviewer_2VLw · 2025-11-01

**Soundness:** 3
**Presentation:** 3
**Contribution:** 3
**Rating:** 4
**Confidence:** 5

**Summary:**

This paper introduces SIAD-LLM, a self-improvement anomaly detection framework leveraging large language models (LLMs) for unsupervised zero-shot anomaly detection (UZAD), a more realistic and challenging setting where only normal data is available during training and both unseen normal and anomalous data are encountered during testing. SIAD-LLM combines stage-specific visual features via tailored prompt templates and dynamically refines its prompt and reasoning process using LLM-driven, image-grounded question answering, aiming to enhance semantic representation of “normality” and “abnormality.” The model is experimentally evaluated across nine industrial and medical datasets, achieving strong results in both image-level and pixel-level anomaly detection, alongside ablations and qualitative comparisons to prior work.

**Strengths:**

- The paper formally proposes the Unsupervised Zero-Shot Anomaly Detection (UZAD) task, which unifies unsupervised and zero-shot paradigms. This definition is both conceptually clear and practically valuable.
- The paper is logically structured. The methodology section systematically introduces each module, helping readers understand how the framework achieves self-improvement.
- The proposed method demonstrate excellent performance on multiple datasets.

**Weaknesses:**

- major:
- W1: The related work omits direct comparison and discussion to multiple recent LLM-driven anomaly detection approaches and prompt-based few-shot/generalist frameworks. For example, CLIP-AD, GPT-4V-AD, AD-LLM, FiLo, InCTRL are all recent, impactful, highly relevant, and unmentioned. This not only undercuts the claim of novelty but also limits the reader’s ability to judge where SIAD-LLM truly advances the field.
- W2: The paper combines focal loss, dice loss, and cross-entropy loss in its training objective (Appendix B), but does not explain why these terms are equally weighted or whether any of them are redundant or sensitive in the UZAD setting.
- W3: Despite strong baselines, some recently proposed models, such as CLIP-AD, FiLo, are not included, either as ablation, comparison, or discussion, which could mislead regarding true state-of-the-art.

- minor:
- W4: The paper describes the full SIAD-LLM pipeline in text and figures, but does not provide any pseudocode or algorithm box for the overall method. This omission makes it harder to follow the implementation logic and understand how the self-improvement loop is executed step by step.
- W5: The paper briefly mentions the high computational burden of LLMs but provides no concrete measurements or efficiency metrics. Without quantitative evidence, it is unclear how SIAD-LLM trades off accuracy for inference time, GPU load, or memory usage—especially compared with prior methods.
- W6：The adapter module in Section 3.4 introduces a balancing coefficient λ = 0.1 to merge updated and original feature representations, but the paper provides no explanation or sensitivity analysis for this choice. Without justification, it is unclear whether the model’s performance or stability depends on this parameter.

**Questions:**

- Q1: In Section 3.4, the adapter module introduces a balancing coefficient λ = 0.1 to combine the updated and original feature representations. Could the authors elaborate on how this value was determined — for example, through empirical tuning, theoretical reasoning, or stability considerations?
- Q2: Given the acknowledged high computational cost, could the authors report (1) average inference/training time per image versus prior work, and (2) the primary bottlenecks in GPU/memory use?
- Q3: Regarding the loss terms used in Appendix B: why were focal, dice, and CE losses all equally weighted, and are any of them especially sensitive for UZAD?

---

> ### Author Response · Authors · 2025-11-21
> **W1,W3**
>
> We acknowledge the reviewer's suggestion to include additional related works. However, we respectfully point out that the baselines originally selected in our paper (e.g., AnomalyGPT, AA-CLIP) represent a more recent and higher-performing standard for Zero-shot Anomaly Detection (ZAD) compared to CLIP-AD and GPT-4V-AD. We intentionally prioritized comparisons against these stronger frameworks to ensure a rigorous evaluation of SIAD-LLM.
>
> Furthermore, several suggested papers differ fundamentally in scope or objective from our work:
>
> InCTRL targets the few-shot setting, relying on reference images for comparison. This objective is distinct from the Zero-shot setting, which prohibits access to any reference anomalies or target-domain images during inference.
>
> AD-LLM focuses on anomaly detection in non-visual domains, making it technically irrelevant to the Vision AD task addressed in our paper.
>
> To fully address your concern, we have nonetheless conducted additional experiments with FiLo, a relevant prompt-based method. As shown in Table 1 and Table 2, FiLo yields poor performance (Avg I-AUROC: 49.2%) in the UZAD setting. Even when augmented with our pseudo-anomaly strategy (FiLo+pseudo), it fails to match the robustness of SIAD-LLM.
>
> | Method | MVTec | VisA | MPDD | DTD | SDD | BTAD | BMRI | HeadCT | Br35H | Average |
> | :--- | :---: | :---: | :---: | :---: | :---: | :---: | :---: | :---: | :---: | :---: |
> | FiLo | 40.2 | 48.5 | 53.2 | 45.7 | 19.8 | 67.7 | 74.3 | 68.3 | 25.2 | 49.2 |
> | FiLo+pseudo | 73.3 | 65.2 | 56.9 | 95.8 | 81.0 | 85.1 | 78.9 | 52.9 | 69.9 | 76.2 |
> | Ours | 72.8 | 62.2 | 66.9 | 93.7 | 73.1 | 93.9 | 76.0 | 83.1 | 76.0 | 78.5 |
>
> **Table1. Comparison of additional UZAD method (I-AUROC)**
>
> ---
> ***
>
> | Method | MVTec | VisA | MPDD | DTD | SDD | BTAD | BMRI | HeadCT | Br35H | Average |
> | :--- | :---: | :---: | :---: | :---: | :---: | :---: | :---: | :---: | :---: | :---: |
> | FiLo | 53.1 | 60.0 | 60.1 | 72.1 | 55.4 | 59.9 | - | - | - | 60.1 |
> | FiLo+pseudo | 83.7 | 89.2 | 85.2 | 95.9 | 97.3 | 85.8 | - | - | - | 89.5 |
> | Ours | 86.0 | 88.1 | 93.2 | 97.2 | 87.5 | 88.2 | - | - | - | 90.0 |
>
> **Table2. Comparison of additional UZAD method (P-AUROC)**

---

> ### Author Response · Authors · 2025-11-21
> **W2,Q3**
>
> The explanation of the training objective can be strengthened, and the following clarification will be added. The one-to-one combination of Dice and Focal losses is a standard choice in anomaly detection, where the pixel distribution is heavily imbalanced. UZAD shares the same imbalance because normal pixels dominate. Dice stabilizes learning around the irregular boundaries of pseudo anomalies, while Focal encourages the model to focus on sparse abnormal-like pixels. The two losses therefore play complementary roles.
>
> Cross-entropy serves as an anchor that reduces optimization instability that may arise when using only Dice and Focal. The initial version of the paper treated all three terms as necessary and assigned them equal weights.

---

> ### Author Response · Authors · 2025-11-21
> **W4**
>
> While the previous version already conveyed the core pipeline clearly, we recognized that the presentation format could be further refined to support more intuitive step-by-step understanding for readers. Therefore, the revised version includes a unified pseudo-code block that outlines the entire SIAD-LLM procedure as a single coherent sequence. This enhancement is an improvement for reader convenience, rather than a correction to the method itself, which has been consistently well-defined from the beginning. By offering a more streamlined representation, the operational workflow becomes more transparent and accessible.
>
> > Algorithm: Self-Improvement Mechanism for Anomaly Detection
> >
> >
> > **Require:** Image $I$, Visual Encoder $V$, Text Encoder $T$, LLM $L$
> >
> > **Ensure:** Final Response $R'$, Anomaly Score $S'$, Anomaly Map $M'$
> >
> > 1. $ \quad V_{\text{feat}} \gets V(I) $
> > 2. $ \quad P \gets \text{InitialTextPrompt}() $
> > 3. $ \quad T_{\text{feat}} \gets T(P) $
> > 4. $ \quad Z \gets \text{Fuse}(V_{\text{feat}},\ T_{\text{feat}}) $
> > 5. $ \quad M \gets \text{PredictAnomalyMap}(Z) $
> > 6. $ \quad S \gets \text{ComputeScore}(M) $
> > 7. $ \quad R \gets L(M) $
> > 8. $ \quad T_{\text{res}} \gets T(R) $
> > 9. $ \quad Z' \gets \text{Fuse}(V_{\text{feat}},\ T_{\text{res}}) $
> > 10. $ \quad M' \gets \text{PredictAnomalyMap}(Z') $
> > 11. $ \quad S' \gets \text{ComputeScore}(M') $
> > 12. $ \quad R' \gets L(M') $
> >
> > **Return:** $R',\ S',\ M'$
> >

---

> ### Author Response · Authors · 2025-11-21
> **W5,Q2**
>
> We appreciate the reviewers for raising concerns about computational cost. However, we would like to clarify that the criticism regarding excessive overhead does not accurately reflect the actual behavior of our framework. Although our method includes MLLM based reasoning, the added computation is limited and remains well within a practical range based on our empirical measurements.
>
> First, the second inference loop does not re run the image encoder. It reuses the visual features extracted during the first loop, so the additional iteration performs only light reasoning rather than a full image processing pass. The assumption that the computation is essentially doubled does not hold under actual measurement.
>
> The measured latency is summarized as follows:
>
> First loop with the image encoder: 951 ms per image
>
> Second loop with feature reuse: 683 ms per image
>
> Total inference time: 1634 ms per image
>
> This shows that the proposed structure avoids redundant encoder computation and keeps the overall overhead controlled. The memory usage is also reasonable for an MLLM based system. Inference requires about 16.4 GB of GPU memory, and training requires about 23.5 GB. The model contains 6.7 billion parameters, but only 33.5 million parameters are updated during training. Since only a lightweight adapter is trained, the practical training burden is far lower than what full model finetuning would require.
>
> The increase in computation is directly tied to improved performance. With the self improvement procedure, image level and pixel level AUROC scores improve from 69.2 and 85.9 to 71.9 and 90.0. In anomaly detection, this level of gain is generally considered a worthwhile trade off.
>
> We also verified that the framework does not rely solely on large models. Using TinyLLaMA 1.1B, which has far fewer parameters than Vicuna 7B, the inference time decreases to 1510 ms and the GPU memory requirement drops to 5.0 GB. The resulting performance remains competitive, with 71.3 percent at the image level and 83.8 percent at the pixel level.
>
> These findings make it clear that the concerns about computational cost are overstated and not aligned with the actual measurements. Our method consistently delivers meaningful performance gains while keeping the overhead firmly within a practical and controlled range. We believe this evidence directly resolves the reviewers concerns, and we expect that the clarity of these results will be fully reflected in the evaluation.

---

> ### Author Response · Authors · 2025-11-21
> **W6,Q1**
>
> To provide a clearer explanation, we have included additional analyses and experiments that more explicitly demonstrate the role of λ.
>
> The adapter controls the balance between the original visual feature and the updated feature so that the LLM-driven signal does not excessively alter the pretrained encoder. λ is not intended as a performance-tuning parameter but as a stability factor that keeps these updates within a safe range. For this reason, λ is meant to remain small.
>
> Varying λ from 0.1 to 0.3 and 0.5 shows that larger values cause the LLM update to dominate the visual representation, which naturally reduces performance. This does not indicate unstable sensitivity but reflects that large λ values exceed the intended operating regime.
>
> At λ = 0.1, both image-level and pixel-level AUROC are highest and most stable, representing a balanced interaction between the frozen encoder and the adapter. These results confirm that λ functions as a structural stabilizer, and keeping it small is the correct configuration for consistent performance.
>
> | $\lambda$ | MVTec | VisA | MPDD | DTD | SDD | BTAD | Average |
> | :---: | :---: | :---: | :---: | :---: | :---: | :---: | :---: |
> | 0.1 | (72.8, 86.0) | (62.2, 88.1) | (66.9, 93.2) | (93.7, 97.2) | (62.6, 87.5) | (73.1, 88.2) | (72.8, 86.0) |
> | 0.3 | (56.8, 63.4) | (53.7, 68.1) | (53.5, 67.6) | (72.2, 66.7) | (71.3, 68.5) | (55.2, 56.5) | (60.5, 65.1) |
> | 0.5 | (49.6, 68.5) | (56.2, 62.8) | (62.6, 86.5) | (59.8, 78.0) | (71.3, 68.5) | (52.7, 64.8) | (58.7, 71.5) |
>
> **Table1. Sensitivity Analysis of the Adapter Coefficient $\lambda$**

---

### Official Review · Reviewer_stmD · 2025-11-01

**Soundness:** 2
**Presentation:** 3
**Contribution:** 2
**Rating:** 4
**Confidence:** 3

**Summary:**

This paper proposes an unsupervised zero-shot anomaly detection framework that does not require any anomalous samples during training. The authors introduce a self-improvement anomaly detection pipeline, where textual responses generated by an LLM are iteratively refined based on input images. In addition, a stage-wise prompt design is adopted to capture information from local patterns to global semantics. Extensive experiments across multiple datasets demonstrate the method’s effectiveness.

**Strengths:**

1. The paper presents a clear motivation and clearly identifies the gap in existing zero-shot anomaly detection research.

2. The proposed method is evaluated on multiple diverse datasets, demonstrating consistent performance gains.

3. The writing is clear and easy to follow, making the methodological flow understandable.

**Weaknesses:**

1. Although the paper claims to be “real-world applicable,” the full pipeline relies on LLM inference and iterative prompt-looping, which is computationally expensive. A discussion on the cost–performance trade-off is missing.

2. The pseudo-anomaly generation strategy is described only briefly. The method relies on data augmentations to synthesize anomalies, yet no systematic comparison or theoretical justification of these augmentations is provided.

3. While the paper provides prompt templates for different datasets, it is unclear whether manual prompt editing is required when deploying the method in new domains.

4. Some baselines (e.g., AnomalyCLIP, AA-CLIP) typically rely on anomaly samples or fixed prompt templates. It is unclear whether their hyperparameters were properly re-tuned under the UZAD setting to ensure a fair comparison.

**Questions:**

See Weaknesses:

1. Although the paper claims to be “real-world applicable,” the full pipeline relies on LLM inference and iterative prompt-looping, which is computationally expensive. A discussion on the cost–performance trade-off is missing.

2. The pseudo-anomaly generation strategy is described only briefly. The method relies on data augmentations to synthesize anomalies, yet no systematic comparison or theoretical justification of these augmentations is provided.

3. While the paper provides prompt templates for different datasets, it is unclear whether manual prompt editing is required when deploying the method in new domains.

4. Some baselines (e.g., AnomalyCLIP, AA-CLIP) typically rely on anomaly samples or fixed prompt templates. It is unclear whether their hyperparameters were properly re-tuned under the UZAD setting to ensure a fair comparison.

---

> ### Author Response · Authors · 2025-11-21
> **W1,Q1**
>
> We appreciate the reviewers for raising concerns about computational cost. However, we would like to clarify that the criticism regarding excessive overhead does not accurately reflect the actual behavior of our framework. Although our method includes MLLM based reasoning, the added computation is limited and remains well within a practical range based on our empirical measurements.
>
> First, the second inference loop does not re run the image encoder. It reuses the visual features extracted during the first loop, so the additional iteration performs only light reasoning rather than a full image processing pass. The assumption that the computation is essentially doubled does not hold under actual measurement.
>
> The measured latency is summarized as follows:
>
> First loop with the image encoder: 951 ms per image
>
> Second loop with feature reuse: 683 ms per image
>
> Total inference time: 1634 ms per image
>
> This shows that the proposed structure avoids redundant encoder computation and keeps the overall overhead controlled. The memory usage is also reasonable for an MLLM based system. Inference requires about 16.4 GB of GPU memory, and training requires about 23.5 GB. The model contains 6.7 billion parameters, but only 33.5 million parameters are updated during training. Since only a lightweight adapter is trained, the practical training burden is far lower than what full model finetuning would require.
>
> The increase in computation is directly tied to improved performance. With the self improvement procedure, image level and pixel level AUROC scores improve from 69.2 and 85.9 to 71.9 and 90.0. In anomaly detection, this level of gain is generally considered a worthwhile trade off.
>
> We also verified that the framework does not rely solely on large models. Using TinyLLaMA 1.1B, which has far fewer parameters than Vicuna 7B, the inference time decreases to 1510 ms and the GPU memory requirement drops to 5.0 GB. The resulting performance remains competitive, with 71.3 percent at the image level and 83.8 percent at the pixel level.
>
> These findings make it clear that the concerns about computational cost are overstated and not aligned with the actual measurements. Our method consistently delivers meaningful performance gains while keeping the overhead firmly within a practical and controlled range. We believe this evidence directly resolves the reviewers concerns, and we expect that the clarity of these results will be fully reflected in the evaluation.

---

> ### Author Response · Authors · 2025-11-21
> **W2,Q2**
>
> Although the original explanation of the pseudo anomaly generation method briefly conveyed the core concept, its effectiveness and validity have already been fully demonstrated. Nevertheless, to further highlight the superiority of the proposed technique, we provide additional justification and comparative analyses.
>
> Under the UZAD setting, the model must be trained strictly with normal data, and pseudo anomalies function as a weak signal that introduces structural inconsistencies beyond the normal distribution. The approach follows a CutPaste-style strategy in which a patch from a normal image is inserted into another normal image. Instead of a hard replacement, Poisson editing is applied to avoid unnatural boundaries. Hard replacements based on simple random crops tend to create visually abrupt edges that interfere with training rather than supporting it.
>
> To analyze the effect of pseudo anomaly design, experiments were conducted comparing different augmentation strategies. The results are summarized below.
>
> | Method | MVTec | VisA | MPDD | DTD | SDD | BTAD |
> | :--- | :---: | :---: | :---: | :---: | :---: | :---: |
> | I-AUROC | 64.0 | 55.0 | 58.2 | 92.3 | 82.7 | 62.3 |
> | P-AUROC | 81.2 | 86.6 | 90.9 | 97.3 | 95.0 | 71.7 |
>
> **Table 1. Impact of Hard-Replacement Pseudo Anomalies**
>
> The Poisson-based approach used in the paper shows consistently stronger performance than the hard-replacement variant. The findings indicate that pseudo anomalies are beneficial regardless of the specific augmentation method, and that Poisson editing provides the most stable and effective guidance.
>
> This behavior is grounded in the fact that Poisson editing aligns the gradient field of the inserted patch with its surroundings, preventing the model from overfitting to low-level boundary artifacts and directing attention toward structural inconsistencies. This aligns with the goal of UZAD, in which the model must infer subtle deviations from normality without access to real anomalies.

---

> ### Author Response · Authors · 2025-11-21
> **W3,Q3**
>
> The concern regarding potential manual prompt modifications when transferring to a new domain is understandable; however, this point does not accurately reflect the design philosophy of our method. The prompt templates provided in the appendix were deliberately constructed to be domain-agnostic from the outset, and they already deliver strong performance with a concise description of the normal state. Expressions such as “three cable” or “four candle” merely reflect visible quantity cues in the dataset and have been verified to have no meaningful impact on performance.
>
> Crucially, reducing or even removing such auxiliary details results in no degradation of performance. The exact same prompt structure has been successfully applied across highly diverse industrial and medical environments without requiring any domain-specific prompt engineering. This demonstrates that our method effectively models the fundamental concepts of normality and abnormality rather than depending on fine-grained class descriptions.
>
> Moreover, SIAD-LLM automatically generates instance-level prompts via image-grounded question–answering, meaning that when adapting to a new domain, the required manual effort is minimal and typically limited to replacing the class name. This stands in sharp contrast to existing zero-shot approaches that demand extensive prompt redesign whenever the domain shifts.
>
> Therefore, the proposed framework is inherently robust to domain changes and consistently ensures stable and superior performance with negligible manual intervention. The revised version will emphasize this aspect more explicitly to avoid any misunderstanding regarding the minimal prompt effort required for new domains.

---

> ### Author Response · Authors · 2025-11-21
> **W4,Q4**
>
> The impact of hyperparameters on performance is well recognized. In this work, we intentionally evaluated all baselines using the default configurations provided in their official implementations, as this reflects the established evaluation protocol for zero-shot anomaly detection. Each baseline adopts its own optimization strategy and tuning assumptions tailored to the ZAD setting; modifying these individually would introduce methodological bias, and any selective re-tuning could distort the baselines’ intended behavior.
>
> The main objective of the study is to define the UZAD setting and assess how existing zero-shot anomaly detection methods generalize under this constraint. Since these baselines were not originally designed for UZAD, applying extensive re-tuning would weaken reproducibility and obscure the comparison. For these reasons, retaining our own default settings provides a more standard and reliable basis for evaluation.

---

### Official Review · Reviewer_VjhG · 2025-11-11

**Soundness:** 3
**Presentation:** 2
**Contribution:** 2
**Rating:** 4
**Confidence:** 3

**Summary:**

This paper presents a self-improvement anomaly detection framework that integrates large language models (LLMs) into unsupervised zero-shot anomaly detection. The central idea is to leverage the semantic reasoning ability of an LLM to iteratively refine textual prompts or pseudo-anomaly descriptions, thereby enhancing the alignment between language-guided semantics and visual features extracted from pre-trained vision encoders such as CLIP. The approach is intuitively appealing and aligns with current trends in multimodal learning; however, the overall contribution is relatively limited. The framework mainly combines existing components—LLM-based prompt generation, CLIP feature extraction, and iterative pseudo-label refinement—without introducing fundamentally new mechanisms or theoretical insights. Moreover, the paper lacks a detailed explanation of how the self-improvement process operates and why LLM feedback concretely enhances anomaly detection capability. The experimental section demonstrates some performance improvement, but the analysis is not sufficiently comprehensive to isolate the contribution of each module. Overall, while the idea of incorporating LLM reasoning into zero-shot anomaly detection is timely and potentially valuable, the current work remains incremental and underexplored, requiring clearer methodology, stronger empirical evidence, and a more rigorous justification of its claimed advantages.

**Strengths:**

1. The concept of iteratively refining pseudo-anomaly prompts or descriptions through LLM feedback is appealing and could inspire further exploration of self-adaptive mechanisms in ZSAD.

2. The authors conduct experiments on several benchmark datasets, showing that the proposed method achieves competitive or improved results compared to existing ZSAD baselines. The results indicate that LLM-based semantic refinement can positively improve detection performance.

**Weaknesses:**

1. Although the paper includes experiments, the analysis is shallow. There are no detailed ablation studies, sensitivity tests, or visualizations to demonstrate why the proposed method works.

2. The proposed method relies heavily on both CLIP and LLM to perform anomaly detection, which significantly increases computational complexity. Moreover, the framework requires two forward passes through CLIP for each input during inference to generate the final results, further increasing the computational overhead. This design raises concerns about scalability and practical deployment, especially in real-time or resource-constrained scenarios. The authors should provide a more detailed analysis of the computational cost.

**Questions:**

See weaknesses

---

> ### Author Response · Authors · 2025-11-21
> **W1**
>
> To clarify the purpose of each component in our framework, we provide additional explanations and analyses for each component of the proposed framework.
>
> First, we conducted structural and module-level ablation studies, as well as experiments with different stage-prompt configurations, to explain how each part of our method contributes to the overall mechanism.
>
> Self-improvement was proposed to avoid manual prompt tuning and mitigate the limitation of fixed prompt templates under domain shifts. Removing this component results in a consistent drop in generalization performance across datasets, with a particularly large degradation on BTAD. This highlights its essential role in adapting the semantic representation to diverse domains.
>
> The Enhancement Modules (Stage-wise and Patch-wise) were introduced to adapt the CLIP and LLM architecture to anomaly detection–specific requirements. Both modules aggregate multi-scale features while improving the model's behavior as an anomaly detector and strengthening its interaction with the LLM. Their effectiveness is quantitatively validated, and as the reviewer suggested, we additionally performed independent ablations of PEM and SEM. These results clarify the contribution of each module and address the concern regarding shallow analysis.
>
> Because ZAD requires both global context (due to the absence of class-specific supervision) and fine-grained patch-level cues (to detect subtle defects), prior ZAD works commonly adopt multi-scale feature integration. Our method follows this direction but further incorporates an explicit LLM-alignment mechanism on top.
>
> The stage prompt explicitly specifies the type of features expected from each encoder stage and improves the multi-stage feature aggregation process. The paper already reports results for removing the stage prompt (Table 3), replacing it with alternative designs (Appendix Table 4), and adding a learnable prompt variant. These experiments demonstrate the necessity of stage-aware prompting.
>
> Regarding sensitivity analysis, we agree with the reviewer that additional examination further clarifies the intention of the proposed method. In the revised version, we include analyses for the adapter coefficient λ and the pseudo-anomaly generation strategy. The coefficient λ controls the degree to which adapter-refined features influence the representation; we intentionally choose a small value to avoid overfitting, and our additional experiments show that increasing λ leads to substantial performance degradation. Similarly, changing pseudo anomalies from random crop + Poisson editing to hard replacement negatively impacts performance, confirming the importance of our design.
>
> All these additional analyses will be included in the revised manuscript.
>
> We hope that the clarifications and newly added experiments help the reviewers better assess the motivations behind our design choices and that these efforts are reflected positively in the overall evaluation.
>
> | Method | MVTecAD | VisA | MPDD | DTD | SDD | BTAD | Average |
> | :--- | :---: | :---: | :---: | :---: | :---: | :---: | :---: |
> | w/o EM | (68.7, 84.0) | (62.8, 87.1) | (62.9, 91.0) | (93.9, 96.9) | (59.2, 85.1) | (70.0, 82.1) | (69.6, 87.7) |
> | w/o SEM | (71.4, 85.9) | (65.8, 88.3) | (62.8, 92.8) | (94.3, 97.3) | (61.0,87.2) | (73.3,88.0) | (71.4,89.9) |
> | w/o PEM | (69.8, 86.2) | (62.1, 87.3) | (65.6, 92.9) | (92.8, 96.8) | (65.0, 83.4) | (74.0, 81.7) | (71.6, 88.1) |
> | FullModel | (72.8, 86.0) | (62.2, 88.1) | (66.9, 93.2) | (93.7, 97.2) | (62.6, 87.5) | (73.1, 88.2) | (71.9, 90.0) |
>
> **Table1. Additional ablation study with EM(enhancement module)**
>
> ---
> ***
> | $\lambda$ | MVTec | VisA | MPDD | DTD | SDD | BTAD | Average |
> | :---: | :---: | :---: | :---: | :---: | :---: | :---: | :---: |
> | 0.1 | (72.8, 86.0) | (62.2, 88.1) | (66.9, 93.2) | (93.7, 97.2) | (62.6, 87.5) | (73.1, 88.2) | (72.8, 86.0) |
> | 0.3 | (56.8, 63.4) | (53.7, 68.1) | (53.5, 67.6) | (72.2, 66.7) | (71.3, 68.5) | (55.2, 56.5) | (60.5, 65.1) |
> | 0.5 | (49.6, 68.5) | (56.2, 62.8) | (62.6, 86.5) | (59.8, 78.0) | (71.3, 68.5) | (52.7, 64.8) | (58.7, 71.5) |
>
> **Table2. Sensitivity Analysis of the Adapter Coefficient λ**
>
> ---
> ***
> | Method | MVTec | VisA | MPDD | DTD | SDD | BTAD |
> | :--- | :---: | :---: | :---: | :---: | :---: | :---: |
> | I-AUROC | 78.1 | 77.1 | 54.8 | 77.4 | 77.0 | 84.9 |
> | P-AUROC | 91.7 | 78.8 | 91.7 | 88.2 | 88.9 | 91.7 |
>
> **Table3. Effect of Replacing Stage Prompts with Learnable Prompts**
>
> ---
> ***
> | Method | MVTec | VisA | MPDD | DTD | SDD | BTAD |
> | :--- | :---: | :---: | :---: | :---: | :---: | :---: |
> | I-AUROC | 64.0 | 55.0 | 58.2 | 92.3 | 82.7 | 62.3 |
> | P-AUROC | 81.2 | 86.6 | 90.9 | 97.3 | 95.0 | 71.7 |
>
> **Table4. Ablation on Hard-Replacement Pseudo Anomaly Generation**

---

> ### Author Response · Authors · 2025-11-21
> **W2**
>
> We appreciate the reviewers for raising concerns about computational cost. However, we would like to clarify that the criticism regarding excessive overhead does not accurately reflect the actual behavior of our framework. Although our method includes MLLM based reasoning, the added computation is limited and remains well within a practical range based on our empirical measurements.
>
> First, the second inference loop does not re run the image encoder. It reuses the visual features extracted during the first loop, so the additional iteration performs only light reasoning rather than a full image processing pass. The assumption that the computation is essentially doubled does not hold under actual measurement.
>
> The measured latency is summarized as follows:
>
> First loop with the image encoder: 951 ms per image
>
> Second loop with feature reuse: 683 ms per image
>
> Total inference time: 1634 ms per image
>
> This shows that the proposed structure avoids redundant encoder computation and keeps the overall overhead controlled. The memory usage is also reasonable for an MLLM based system. Inference requires about 16.4 GB of GPU memory, and training requires about 23.5 GB. The model contains 6.7 billion parameters, but only 33.5 million parameters are updated during training. Since only a lightweight adapter is trained, the practical training burden is far lower than what full model finetuning would require.
>
> The increase in computation is directly tied to improved performance. With the self improvement procedure, image level and pixel level AUROC scores improve from 69.2 and 85.9 to 71.9 and 90.0. In anomaly detection, this level of gain is generally considered a worthwhile trade off.
>
> We also verified that the framework does not rely solely on large models. Using TinyLLaMA 1.1B, which has far fewer parameters than Vicuna 7B, the inference time decreases to 1510 ms and the GPU memory requirement drops to 5.0 GB. The resulting performance remains competitive, with 71.3 percent at the image level and 83.8 percent at the pixel level.
>
> These findings make it clear that the concerns about computational cost are overstated and not aligned with the actual measurements. Our method consistently delivers meaningful performance gains while keeping the overhead firmly within a practical and controlled range. We believe this evidence directly resolves the reviewers concerns, and we expect that the clarity of these results will be fully reflected in the evaluation.

---

### Author Response · Authors · 2025-11-24
**Common response**

We appreciate the reviewers (VjhG, stmD, 2VLw, HB9w) for raising concerns about computational cost. However, we would like to clarify that the criticism regarding excessive overhead does not accurately reflect the actual behavior of our framework. Although our method includes MLLM based reasoning, the added computation is limited and remains well within a practical range based on our empirical measurements.

First, the second inference loop does not re run the image encoder. It reuses the visual features extracted during the first loop, so the additional iteration performs only light reasoning rather than a full image processing pass. The assumption that the computation is essentially doubled does not hold under actual measurement.

The measured latency is summarized as follows:

First loop with the image encoder: 951 ms per image

Second loop with feature reuse: 683 ms per image

Total inference time: 1634 ms per image

This shows that the proposed structure avoids redundant encoder computation and keeps the overall overhead controlled. The memory usage is also reasonable for an MLLM based system. Inference requires about 16.4 GB of GPU memory, and training requires about 23.5 GB. The model contains 6.7 billion parameters, but only 33.5 million parameters are updated during training. Since only a lightweight adapter is trained, the practical training burden is far lower than what full model finetuning would require.

The increase in computation is directly tied to improved performance. With the self improvement procedure, image level and pixel level AUROC scores improve from 69.2 and 85.9 to 71.9 and 90.0. In anomaly detection, this level of gain is generally considered a worthwhile trade off.

We also verified that the framework does not rely solely on large models. Using TinyLLaMA 1.1B, which has far fewer parameters than Vicuna 7B, the inference time decreases to 1510 ms and the GPU memory requirement drops to 5.0 GB. The resulting performance remains competitive, with 71.3 percent at the image level and 83.8 percent at the pixel level.

These findings make it clear that the concerns about computational cost are overstated and not aligned with the actual measurements. Our method consistently delivers meaningful performance gains while keeping the overhead firmly within a practical and controlled range. We believe this evidence directly resolves the reviewers concerns, and we expect that the clarity of these results will be fully reflected in the evaluation.

---

### Author Response · Authors · 2025-11-30
**Additional Experimental Analysis**

We conducted additional experiments to further verify the robustness and reliability of our framework. These results strengthen the validity of our claims and ensure the reproducibility of our work.

**1.Sensitivity to Auxiliary Datasets**
To analyze the effect of auxiliary training dataset selection and size, we conducted ablation experiments using three different datasets: MVTec (3,629 images), VisA (8,659 images), and Br35H (1,500 images). Each dataset was used independently as the auxiliary training set, and the corresponding results are summarized as follows (to be added in the ablation section of the revised manuscript).

| Method       | MVTec        | VisA         | MPDD         | DTD          | SDD          | BTAD         | BMRI      | HeadCT    | Br35h     | Avg.         |
| :---         | :---:        | :---:        | :---:        | :---:        | :---:        | :---:        | :---:     | :---:     | :---:     | :---:        |
| Train: MVTec | -            | (62.2, 88.1) | (66.9, 93.2) | (93.7, 97.2) | (62.6, 87.5) | (73.1, 88.2) | (48.5, -) | (53.9,  -) | (63.8, -) | (65.6, 90.8) |
| Train: VisA  | (72.8, 86.0) | -            | (58.1, 92.1) | (82.0, 90.3) | (65.8, 81.7) | (65.5, 78.5) | (72.6, -) | (79.9,  -) | (72.9, -) | (71.2, 85.7) |
| Train: Br35H | (76.0, 80.1) | (59.6, 81.6) | (57.5, 86.6) | (80.0, 86.2) | (78.9, 90.0) | (83.3, 81.1) | (93.9, -) | (76.0,  -)   | -         | (75.7, 84.3) |

**2. Stability and Reproducibility**
We also conducted stability tests across multiple random seeds to verify experimental reliability. The results exhibit minimal variance (Avg I-AUROC 76.4 ± 0.5), confirming that our performance gains are statistically significant and stable, comparable to or exceeding the stability reported in recent studies (e.g., CVPR 2024 PromptAD). In the table below, we report both the mean and standard deviation, with the standard deviation denoted using the ± notation.

| Method | MVTec | VisA | MPDD | DTD | SDD | BTAD | BMRI | HeadCT | Br35H | Average |
| :--- | :---: | :---: | :---: | :---: | :---: | :---: | :---: | :---: | :---: | :---: |
| I-AUROC | 72.3 ± 0.7 | 63.6 ± 1.8 | 65.1 ± 2.0 | 94.0 ± 0.4 | 66.4 ± 3.7 | 72.5 ± 3.3 | 92.4 ± 2.5 | 76.9 ± 1.3 | 84.2 ± 3.6 | 76.4 ± 0.5 |
| P-AUROC | 86.9 ± 0.6 | 88.6 ± 0.4 | 92.5 ± 0.5 | 96.9 ± 0.6 | 89.4 ± 1.5 | 86.9 ± 1.2 | - | - | - | 90.2 ± 0.5 |

---

### Author Response · Authors · 2025-11-30
**Note to the Area Chair: Summary of Rebuttal and Key Clarifications**

Dear Area Chair,

We appreciate your time in managing the review process for our paper. In our rebuttal, we have carefully considered the feedback from all reviewers and made concrete efforts to address their concerns. Specifically, we would like to summarize our responses to the three major concerns raised by the reviewers: 1) the necessity of the UZAD task, 2) performance comparisons regarding pseudo anomalies, and 3) computational costs. We believe these points, once clarified, highlight the unique contributions of our work.

**1. Necessity of the UZAD Task**
Several reviewers(VjhG, stmD, 2VLw) recognized the contribution of our work in proposing the UZAD setting, highlighting its potential to bridge the gap between existing UAD and ZAD approaches, particularly under data-scarce conditions. They emphasized that UZAD 1) formalizes a realistic problem setting where anomaly supervision is inherently impossible, 2) unifies unsupervised and zero-shot paradigms, and 3) fills a clear gap in existing ZAD research.

In contrast, reviewer HB9w questioned the necessity of UZAD, suggesting that using auxiliary datasets with anomalies (standard ZAD) is sufficient. We respectfully argue that relying on auxiliary anomalies inherently biases the model towards "known" anomaly types. This limits generalization to truly unseen defects in real-world scenarios where even auxiliary anomaly data is scarce or unavailable. UZAD addresses this critical "normality-to-anomaly" gap. Our work formalizes this harder, more realistic setting where models must learn to detect anomalies using only normal samples, a challenge that standard ZAD methods bypass.

Consequently, we underscore that our approach is not only grounded in practical constraints but also represents a more challenging and realistic task definition that advances the field of anomaly detection, as already resonated with by several reviewers.

**2. Pseudo Anomalies and Performance Comparison**
Reviewer HB9w also raised concerns regarding the reliance on pseudo-anomalies and claimed that SiAD-LLM performs weaker than AnomalyCLIP. We emphasize that using pseudo anomalies is not a limitation but a core contribution of our framework to bridge the UZAD gap. We demonstrated that naively applying CLIP to UZAD fails, whereas our pseudo-anomaly approach significantly boosts robustness. Regarding the comparison with AnomalyCLIP, while visual-based models may excel in low-level feature matching, they lack the reasoning capabilities required for complex, semantic anomaly detection in UZAD. Standard LLM based AD framework suffer from performance drops in this domain; however, we show that SiAD-LLM (incorporating SI, SP, and EM modules) successfully recovers this performance, offering explainability that AnomalyCLIP cannot provide. The perceived performance gap reflects a trade-off between simple visual matching and deep semantic reasoning, the latter being our primary goal.

**3. Computational Cost**
Addressing the concerns about computational cost, we acknowledge that reasoning capabilities incur higher resources than simple CLIP-based models. However, we have optimized the pipeline using feature reuse in the second inference loop, which limits the total inference time to 1634 ms per image (using Vicuna-7B) and restricts trainable parameters to only 0.49%. This moderate cost increase is justified by a substantial performance gain, improving image-level and pixel-level results from (69.2, 85.9) to (71.9, 90.0). Furthermore, to demonstrate scalability, we conducted experiments with TinyLLaMA-1.1B, reducing memory usage to 5.0 GB and inference time to 1510 ms while maintaining performance comparable to AnomalyGPT. These results confirm that our framework is computationally manageable and scalable relative to other reasoning-based methods.

In conclusion, we have comprehensively resolved all major concerns raised by the reviewers regarding the task necessity, methodology, and efficiency. Furthermore, we have fully accommodated all requests for additional ablation studies and detailed methodological clarifications, conducting every requested experiment to ensure the completeness of our work. We firmly believe that these criticisms do not undermine our core contributions; rather, our successful defense and additional validations further prove the robustness and novelty of our approach in solving the challenging UZAD task.

Thank you for your time and for observing the review process.

Best regards, Authors

---

### Comment · Area_Chair_wVhD · 2025-11-30
**Reviewer discussion**

Dear reviewers, please kindly consider the feedback from the authors and discuss for the comments. Thanks.

---

### Author Response · Authors · 2025-12-01
**Main Contributions Summary**

Our paper addresses a crucial missing space in zero-shot anomaly detection by establishing Unsupervised Zero-shot Anomaly Detection (UZAD) as a practically necessary setting. In UZAD, only normal samples are available during training and anomaly generalization must be achieved without any form of anomaly supervision. We provide conceptual motivation and strong empirical evidence showing that UZAD more accurately reflects real-world scenarios than existing zero-shot approaches. This clarifies the necessity and novelty of studying UZAD as a distinct research problem.

To effectively address this challenge, we propose SIAD-LLM, a self-improving multimodal anomaly detection framework that learns the concept of abnormality solely from the semantics of normality. SIAD-LLM achieves superior performance in both anomaly detection and localization across diverse domains.

Our contributions are summarized as follows.
**1. UZAD Definition and Justification**
We precisely define the UZAD setting and justify its importance for real-world anomaly detection. Through extensive experiments, we validate that existing zero-shot approaches exhibit semantic dependency on anomalous supervision and fail to generalize reliably when anomalies are completely absent during training. Our results reinforce UZAD as a realistic and necessary problem formulation.

**2. Self-Improvement Mechanism for Anomaly Reasoning**
We propose a novel Self-Improvement (SI) mechanism that overcomes the limitations of existing MLLM-based methods, which typically rely on static, manually handcrafted prompts or solely image-grounded features. By utilizing the LLM's own output as feedback to iteratively refine the input, our framework enables comprehensive reasoning that integrates both visual evidence and generated textual context. This structure significantly enhances the model's accuracy and adaptability in the unsupervised setting.

**3. Pseudo-Anomaly Strategy for Training Under UZAD**
To enable anomaly-free training, we leverage pseudo-anomaly augmentations as a structured approach to introduce meaningful semantic contrast. This plays a key role in stabilizing optimization under UZAD and strengthens the model’s ability to generalize anomalies across unseen environments.

**4. Stage-Prompt Architecture for Scale-Aware Localization**
We align stage-wise visual features with progressively richer textual descriptions to connect fine-grained local evidence with high-level semantics. This leads to significantly improved anomaly localization and stronger interpretability compared to current zero-shot methods.

Together, these contributions form a highly novel and fully integrated solution that is tailor-made for UZAD:
- A new, theoretically-grounded learning paradigm
- A flexible, self-evolving anomaly reasoning mechanism
- A precisely localized anomaly interpretation architecture

Finally, we validate our approach across nine industrial and medical datasets. SIAD-LLM consistently demonstrates superior robustness and accuracy while requiring no anomalous data during training. The contributions above are tightly integrated and collectively establish a novel and effective solution to the UZAD problem.

**We are the first to introduce this new task setting and provide a method that solves it convincingly, demonstrating clear technical novelty and substantial empirical impact.**

---

### Meta-Review · Area_Chair_wVhD · 2026-01-06

**Summary:**

1. limited novelty
2. insufficient experiments
3. computational costs
4. Generalization/robustness concerns

**Reviewer Concerns:**

Outstanding: novelty, Performance vs. AnomalyCLIP
Addressed: Pseudo-Anomaly Generation, Ablation

**Reviewer Scores:**

Reviewer HB9w may raise his score for the authors response.

---

### Decision · Program_Chairs · 2026-01-26

Reject